# Cross-Modal Alignment via Variational Copula Modelling

Feng Wu [* 1]  Tsai Hor Chan [* 1]  Fuying Wang [1]  Guosheng Yin [1]  Lequan Yu [1]

## Abstract

Various data modalities are common in real-world applications (e.g., electronic health records, medical images and clinical notes in healthcare). It is essential to develop multimodal learning methods to aggregate various information from multiple modalities. The main challenge is how to appropriately align and fuse the representations of different modalities into a joint distribution. Existing methods mainly rely on concatenation or the Kronecker product, oversimplifying the interaction structure between modalities and indicating a need to model more complex interactions. Additionally, the joint distribution of latent representations with higher-order interactions is underexplored. Copula is a powerful statistical structure for modelling the interactions among variables, as it naturally bridges the joint distribution and marginal distributions of multiple variables. We propose a novel copula-driven multimodal learning framework, which focuses on learning the joint distribution of various modalities to capture the complex interactions among them. The key idea is to interpret the copula model as a tool to align the marginal distributions of the modalities efficiently. By assuming a Gaussian mixture distribution for each modality and a copula model on the joint distribution, our model can generate accurate representations for missing modalities. Extensive experiments on public MIMIC datasets demonstrate the superior performance of our model over other competitors. The code is available at https://github.com/HKU-MedAI/CMCM.

## 1. Introduction

Multimodal learning aims to aggregate information from multiple modalities to generate meaningful representations for downstream tasks. It has been widely explored in the context of vision-language models (Fu et al., 2023; El Banani et al., 2023), audio-visual applications (Chen et al., 2023; Mo & Tian, 2023; Huang et al., 2023), image-video models (Girdhar et al., 2023; Gan et al., 2023) and healthcare applications (Wu et al., 2024a; Hayat et al., 2022). For example, multimodal learning has been applied to various healthcare tasks such as clinical prediction tasks (Zhang et al., 2023; Wu et al., 2024a), report generation (Song et al., 2022; Cao et al., 2023), and clinical trial site selection (Theodorou et al., 2024). The existing fusion strategies can be divided into early, joint, or late fusion (Huang et al., 2020), where the joint fusion paradigm is the most popular strategy and its core idea is to model the interactions between the representations of the input modalities (Hayat et al., 2022). The resulting fused embedding encodes the structural interaction between the modalities, enabling accurate prediction for each modality.

However, due to the heterogeneity of different modalities (e.g., electronic health records: EHRs, medical images, medical reports), properly aligning their distributions remains a challenge. The existing alignment strategies mainly rely on concatenation or Kronecker products which oversimplify the interactions among different modalities. A recent work (Salzmann et al., 2022) emphasizes simple probabilistic assumptions on the marginals and neglects to explore statistical assumptions about the joint distributions of the modalities. This approach may result in biased fused representations, limiting the performance of downstream tasks and the generalizability and robustness of the resulting multimodal models. Therefore, there is still a need for an approach that can more appropriately align the distributions of modalities and model the potentially complex interactions among them.

Copula models have shown great success in modelling the interactions of variables as they construct a bridge between the joint distribution and their marginals (Cherubini, 2004). However, copula models are less explored in deep learning field as most existing approaches heavily rely on sampling-based methods (e.g., MCMC (Silva & Gramacy, 2009)),

[*]Equal contribution  [1]School of Computing and Data Science, University of Hong Kong, Hong Kong, China. Correspondence to: Lequan Yu <lqyu@hku.hk>.

*Proceedings of the 42nd International Conference on Machine Learning*, Vancouver, Canada. PMLR 267, 2025. Copyright 2025 by the author(s).

which are relatively slow and difficult to scale to modern deep learning settings (Smith & Loaiza-Maya, 2023). Although some recent works have attempted to introduce copula to deep learning models through stochastic variational inference (Smith & Loaiza-Maya, 2023), the potential of copula in multimodal learning is still underexplored.

Moreover, existing multimodal learning methods mostly assume the existence of all modalities. In reality, some modalities may be missing for some observations due to various reasons (e.g., missing medical images or reports for some patients due to clinical and administrative factors in healthcare), which may pose a major challenge in multimodal learning. The existing solutions either discard these observations or impute simple values (e.g., zeros or means from other observations) to address the missing modality problem. However, these approaches ignore the marginal distributions of the modality and often mislead the learning of the joint distribution. Therefore, properly learning the marginal distributions is also necessary to generate unbiased latent representations for the observations with missing modalities.

In light of the aforementioned challenges, we propose a novel copula-driven multimodal learning framework, namely $\text{CM}^2$ (**C**ross-**M**odal alignment via variational **C**opula **M**odelling), to tackle the joint fusion paradigm from a probabilistic perspective. Our contributions can be summarized as: (1) We for the first time introduce copula modelling into multimodal learning, where we interpret copula as an effective tool of distribution alignment, guaranteed by Sklar's theorem. (2) We employ a Gaussian mixture model on the marginal distribution of each modality to enable more flexible modelling of the high-dimensional feature distribution of different modalities. (3) We adopt stochastic variational inference to optimize the copula model, which enables the scalability of our model to large-scale datasets. (4) We adopt the learned marginal distribution as the data generator to accurately impute the missing observations. (5) Empirical results on real multimodal MIMIC datasets demonstrate the good performance of our method and ablation analysis corroborates the effectiveness of copula in modality alignments and robustness to potential variations.

## 2. Related Works

**Multimodal Representation Learning.** Multimodal representation learning aims to effectively integrate information from different modalities for accurate predictions on the downstream tasks. Early works (Hayat et al., 2022; Ding et al., 2022; Trong et al., 2020) focus on late fusion that merges unimodal representations via, for instance, concatenation or the Kronecker product. However, such approaches oversimplify the interactions of the modalities and mostly lead to biased fused representations. Therefore, the structural interactions of the modalities need to be encoded in the fused representation for more effective multimodal learning. Recently, modelling the interaction between modalities has received increasing attention. Liang et al. (2024) proposed an information decomposition framework to define and quantify different types of interactions between modalities. Transformer-based methods have greatly facilitated the progress by modelling the cross-model tokens (Zhang et al., 2023; Theodorou et al., 2024). However, matching the correspondence with transformers introduces high computational complexity, which prompts a more efficient approach for representation alignment.

**Copula Deep Learning.** Copula is a promising tool in modelling the interactions or correlations between variables and it constructs a bridge between the joint distribution and marginal distributions. Copula has been widely applied in financial risk management (Hofert, 2021; Rodriguez, 2007), signal processing, and healthcare (Zeng & Wang, 2022) due to its capability in modelling complex interactions. Traditional copula models rely on closed-form solutions of the likelihood and estimate the copula parameter with sampling-based approaches (e.g., MCMC (Silva & Gramacy, 2009)). However, these algorithms suffer from high time complexity, making them less applicable to high-dimensional data. Recently, with the emergence of deep learning, there have been works integrating copula models into deep learning frameworks (Tagasovska et al., 2019; Smith et al., 2020). To tackle the inherent high dimensionality, variational inference is adopted to solve copula models in high dimensions (Tran et al., 2015; Smith & Loaiza-Maya, 2023). For example, Tagasovska et al. (2019) introduced copula to variational autoencoders to create deep generative models. However, the potential of copula in multimodal learning is still underexplored.

**Learning with Missing Data.** Traditional multimodal learning assumes all modalities are available, but in reality, some observations may be missing, like medical images or reports in clinical data. Late fusion is a common strategy to address missing modalities by aggregating predictions (Yoo et al., 2019) or latent space representations (Theodorou et al., 2024) from the available modalities. While effective, it treats each modality independently and lacks interactions among them. Some research focuses on extracting shared information across modalities for downstream tasks (Deldari et al., 2023; Yao et al., 2024), which can be challenging, particularly with heterogeneous modalities like EHRs and CXRs. Under the missing at random (MAR) assumption, imputation methods have become a popular approach for handling missing data. Some approaches assume that the missing modality follows a certain distribution, imputing the missing values using the mean or mode of that distribution (Ma et al., 2021). Others impute missing modalities'

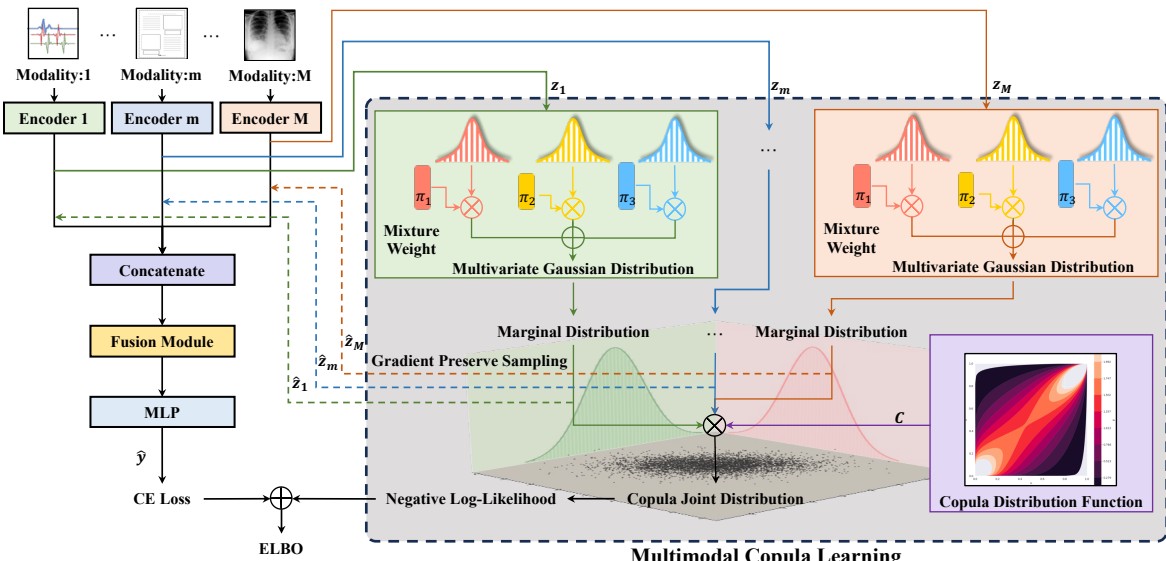

*Figure 1.* Overview of our proposed $\mathtt{CM}^2$ framework. For a dataset with $M$ modalities, we extract modality-specific embeddings $\boldsymbol{z}_m$ via Encoder$_m$ and compute its Gaussian mixture model (GMM). We then model the marginal distributions and estimate the joint distribution using a copula family $C$. We sample $\hat{\boldsymbol{z}}_m$ from its GMM if modality $m$ is missing. The concatenated embedding $\boldsymbol{z}$ then passes through a 2-layer LSTM fusion module and MLP classifier to predict $\hat{\boldsymbol{y}}$. The ELBO for backpropagation can be obtained by aggregating the task-specific loss (e.g., cross-entropy loss) and the negative log-likelihood from the joint distribution.

representations in the latent feature space via deep learning models, attempting to preserve model performance by modeling relationships (Zhang et al., 2022; Wu et al., 2024b) or generating global representations for the missing data (Hayat et al., 2022). Despite their successes, these distributional assumptions or the learned relationships may be inaccurate, potentially introducing bias into the model. Therefore a probabilistic assumption is needed to guarantee the unbiasedness of learned marginal distributions.

## 3. Methodology

### 3.1. Preliminaries

**Copula.** An $M$-variate function $C(u_1, \ldots, c_M)$, where $u_m \in [0, 1]$ for all $m$, is a copula if and only if $C$ defines a valid joint cumulative distribution function (CDF) of the random vector $(U_1, \ldots, U_M)$ with each $U_m$ distributed as uniform on the unit interval. Taking the bivariate Gumbel copula as an example, given the CDF values of the first and second modalities $u$ and $v$, the bivariate distribution is

$$C(u, v; \alpha) = \exp\{-[(-\log u)^\alpha + (-\log v)^\alpha]^{\frac{1}{\alpha}}\}$$

and its copula density is

$$c(u, v; \alpha) = \frac{1}{uv}(-\log v)^{\alpha-1}(-\log u)^{\alpha-1}C(u, v; \alpha)$$
$$\times [g(u, v; \alpha)]^{\frac{2(1-\alpha)}{\alpha}}\left[(\alpha-1)[g(u, v; \alpha)]^{-\frac{1}{\alpha}} + 1\right],$$

where $g(u, v; \alpha) = (-\log u)^\alpha + (-\log v)^\alpha$. The effects of different copula families are discussed in the ablation analysis. Details of different copula families and their corresponding distribution and density functions are provided in Appendix C.

**Multimodal Learning.** Given the multimodal training dataset $\mathcal{D}_{\mathrm{tr}} = \{(\boldsymbol{x}_1^{(i)}, \ldots, \boldsymbol{x}_M^{(i)}, y^{(i)})\}_{i=1}^n$, where $\boldsymbol{x}_m^{(i)}$ is the $i$-th observation of the $m$-th modality and $y^{(i)}$ is the corresponding label, the goal is to train a multimodal model $\mathcal{M}_\Theta(\cdot)$ with parameter $\Theta$ such that the model can achieve optimal performance in downstream tasks.

### 3.2. Copula Multimodal Learning

The overview of the proposed copula-driven multimodal learning framework is shown in Figure 1. Given multimodal data, we extract each modality-specific embedding and compute its Gaussian mixture model (GMM). We then model the marginal densities and estimate the joint distribution using a copula family $C$. If modality $m$ is missing, we generate feature embeddings from its GMM. The concatenated embeddings $\boldsymbol{z}$ are passed through a fusion module and an MLP classifier for prediction. The evidence lower bound (ELBO) combines the copula log-likelihood and task-specific loss.

**Gaussian Mixture Assumption.** The GMM is a common technique in machine learning to model the behavior of distributions in high dimensions (Song et al., 2024; Bai

**Algorithm 1** Sampling algorithm of our proposed method.

1: **Input:**
2: Multimodal model $\mathcal{M}_\Theta(\cdot)$ with parameter $\Theta$
3: The copula parameter $\alpha$
4: Means and covariances of GMM: $\{(\boldsymbol{\mu}_{mk}, \boldsymbol{\Sigma}_{mk}) \mid m = 1, \ldots, M, k = 1, \ldots, K\}$
5: Training set $\mathcal{D}_{\mathrm{tr}} = \{(\boldsymbol{x}_1^{(i)}, \ldots, \boldsymbol{x}_M^{(i)}, y^{(i)})\}_{i=1}^n$
6: **Output:** Trained $f_\Theta$
7: **for** $(\boldsymbol{x}_1^{(i)}, \ldots, \boldsymbol{x}_M^{(i)}, \boldsymbol{y}^{(i)})$ in $\mathcal{D}_{\mathrm{tr}}$ **do**
8: $\quad \hat{y}^{(i)} = \mathcal{M}_\Theta(\boldsymbol{x}_1^{(i)}, \ldots, \boldsymbol{x}_M^{(i)})$
9: $\quad$ Compute task-specific loss $\mathcal{L}_{\mathrm{obj}}$ with $\hat{y}^{(i)}$ and $y^{(i)}$
10: $\quad$ Compute the $\mathrm{KL}(q\|\pi)$ and hence the ELBO
11: $\quad$ Backpropagate the ELBO to update $\Theta, \alpha$
12: **end for**
13: **Return:** Trained $\mathcal{M}_\Theta$

---

et al., 2022; Ni et al., 2021). To generate a more flexible feature distribution, we assume the feature distribution of the $m$-th modality follows a $K$-mixture of multivariate GMM,

$$f_m(\boldsymbol{z}_m) = \sum_{k=1}^{K} \pi_{mk} \mathcal{N}(\boldsymbol{\mu}_{mk}, \boldsymbol{\Sigma}_{mk}), \tag{1}$$

where $\pi_{mk}$ is the mixture weight, $\boldsymbol{\mu}_{mk}$ is the mean vector, and $\boldsymbol{\Sigma}_{mk}$ is the covariance matrix of the $k$-th mixture of the $m$-th modality. Let $\boldsymbol{\mu} = \{\boldsymbol{\mu}_{mk} : m \in [M], k \in [K]\}$ and $\boldsymbol{\Sigma} = \{\boldsymbol{\Sigma}_{mk} : m \in [M], k \in [K]\}$. Without loss of generality, we predict $\pi_{mk}$ with a multilayer perceptron (MLP) with a softmax output layer and adopt the reparameterization trick (Nalisnick, 2018; Tran et al., 2022), which assumes $\boldsymbol{\Sigma}_{mk}$ is diagonal. We further set $\boldsymbol{\mu}$ and $\boldsymbol{\Sigma}$ to be trainable by gradient backpropagation. We compute the cumulative distribution function of the multivariate Gaussian distributions using the approximation provided in Marmin et al. (2015). By employing a mixture model, we can model a wider range of distributions of each modality and improve the flexibility and robustness.

**Multivariate Copula.** Using the multivariate copula, the joint distribution function of the modalities is given by

$$F_{\boldsymbol{z}_1, \ldots, \boldsymbol{z}_M}(\boldsymbol{z}) = C(F_1(\boldsymbol{z}_1), \ldots, F_M(\boldsymbol{z}_M)),$$

where $C(F_1(\boldsymbol{z}_1), \ldots, F_M(\boldsymbol{z}_M))$ is the $M$-dimensional copula distribution function, and $F_m(\boldsymbol{z}_m)$ is the marginal cumulative distribution function of the $m$-th modality which is the CDF of the GMM model defined in Eq. (1).

### 3.3. Stochastic Variational Inference

To tackle the scalability of $\mathrm{CM}^2$ to modern deep learning settings, we adopt the stochastic variational inference to optimize the proposed copula model and treat the copula

parameter $\alpha$ as trainable. Algorithm 1 presents the overall workflow of our method.

**Variational Family.** We use a variational posterior $q$ to approximate the true posterior of the joint distribution. The variational family of the copula model that we optimize during training is given by

$$q(\boldsymbol{z}) = \left[\prod_{m=1}^{M} q_m(\boldsymbol{z}_m)\right] c(Q_1(\boldsymbol{z}_1), \ldots, Q_M(\boldsymbol{z}_M)),$$

where $q_m(\boldsymbol{z}_m)$ is the density of the variational posterior of the GMM of the $m$-th modality, and $Q_m(\boldsymbol{z}_m)$ is the corresponding CDF.

**The Evidence Lower Bound (ELBO).** The joint objective function can be written as the negation of the negative log-likelihood,

$$\begin{aligned} \mathrm{ELBO} = &- \lambda_{\mathrm{cop}} \sum_{i=1}^{n} \Big( \log c(Q_1(\boldsymbol{z}_1^{(i)}), \ldots, Q_M(\boldsymbol{z}_M^{(i)})) \\ &- \sum_{m=1}^{M} \log f_m(\boldsymbol{z}_m^{(i)}) \Big) + \mathcal{L}_{\mathrm{obj}}, \end{aligned}$$

where $f_m(\boldsymbol{z}_m^{(i)})$ is the marginal density of modality $m$, $c(Q_1(\boldsymbol{z}_1^{(i)}), \ldots, Q_M(\boldsymbol{z}_M^{(i)}))$ is the copula density, $\lambda_{\mathrm{cop}}$ is the regularization parameter of the copula, and $\mathcal{L}_{\mathrm{obj}}$ is the task-specific loss (e.g., cross-entropy loss). We compute the gradient based on the ELBO and backpropagate it to $\boldsymbol{\mu}$ and $\boldsymbol{\Sigma}$ to learn the marginal distributions of each modality, with the copula parameter $\alpha$ to learn the interactions among these modalities and the multimodal model parameter $\Theta$ to learn the embedding, fusion, and classification layers.

### 3.4. Handling Missing Modality

Owing to the probabilistic design of our method, our framework can also generate pseudo representations for missing modalities. Without loss of generality, we assume that the missing modalities are missing at random (MAR) and, following prior works (Tran et al., 2017; Ma et al., 2021; Zhang et al., 2022; Wang et al., 2023), we impute the features of these missing modalities in the latent space. We consider missing modalities with complete labels where only the observations are missing. The learned GMM for each modality can be treated as a data generation model, and we can generate feature embeddings through sampling from the GMM of each modality (i.e., $\boldsymbol{z}_m^{(i)} \sim F_m$). Then the generated feature embeddings can be treated as the feature input to the classification layer and predictions can be obtained.

By learning the copula parameter $\alpha$, the marginal distribution of each modality contains information from other modalities and information of the interactions. The generated feature representation $\boldsymbol{z}_m^{(i)}$ can thus better reflect the

characteristics of the joint distribution, which, as a result, can improve the quality of the representation and the downstream task performance.

### 3.5. Theoretical Guarantee with Sklar's Theorem.

We make use of Sklar's theorem to demonstrate the uniqueness of the joint distribution as follows.

**Theorem 3.1.** *(Sklar's theorem) (Sklar, 1959) Let* $F(x_1, \ldots, x_M)$ *be an $M$-variate CDF for* $(X_1, \ldots, X_M)$ *with the marginal CDF for the $m$-th variable given by* $F_m(x_m), m = 1, \ldots, M$.

1. *There exists an $M$-dimensional copula such that*

$$C(F_1(x_1), \ldots, F_M(x_M)) = F(x_1, \ldots, x_M) \quad (2)$$

   *for all $x_m \in \mathbb{R}$.*

2. *Conversely, given any copula $C$ and univariate CDFs* $F_1, \ldots, F_M$, $C$ *is a valid joint CDF for* $(X_1, \ldots, X_M)$. *If $F$ is continuous, then $C$ in Eq. (2) is unique.*

The above theorem lays the foundation to construct joint distributions with the same marginals but different dependence structures, or conversely by fixing the dependence structure and varying the behaviour in individual modalities (Tagasovska et al., 2019). This allows us to update the marginal distributions and the copula parameter separately. Furthermore, since we assume a GMM for each modality and they are continuous by definition, the uniqueness of the copula $C$ can be guaranteed and the identifiability of the model can be enhanced.

## 4. Experiments

### 4.1. Datasets and Experimental Setting

**Datasets.** We evaluate the performance of CM$^2$ using large-scale, real-world EHR datasets: MIMIC-III (Johnson et al., 2016), MIMIC-IV (Johnson et al., 2023), and MIMIC-CXR (Johnson et al., 2019). MIMIC-III and MIMIC-IV are publicly available datasets containing real-world EHR data from patients admitted to the intensive care units (ICUs) or emergency departments of Beth Israel Deaconess Medical Center (BIDMC), comprising numerical time series and clinical notes. MIMIC-CXR is a dataset of Chest X-ray (CXR) images along with radiology reports collected from BIDMC, with a subset of patients matched to those in MIMIC-IV.

Following Hayat et al. (2022), we utilize the MIMIC-IV and MIMIC-CXR datasets for our multimodal experiments. Additionally, we extend our experiments to the MIMIC-III dataset. As CXR images are not available in MIMIC-III, we replace them with clinical notes. Table

*Table 1.* Numbers of samples in training/validation/testing sets

| Datasets | Train | Valid | Test | Total |
|---|---|---|---|---|
| | *Complete Datasets* | | | |
| **MIMIC-III** | 14,681 | 3,222 | 3,236 | 21,139 |
| **MIMIC-III NOTE** | 3,652 | 815 | 806 | 5,273 |
| **MIMIC-IV** | 18,064 | 2,035 | 4,972 | 25,071 |
| **MIMIC-CXR** | 344,529 | 9,497 | 23,069 | 377,095 |
| | *Matched Datasets* | | | |
| **MIMIC-III | NOTE** | 3,652 | 815 | 806 | 5,273 |
| **MIMIC-IV | CXR** | 4,287 | 465 | 1,179 | 5,931 |
| **MIMIC-IV | CXR | REPORT** | 4,287 | 465 | 1,179 | 5,931 |

1 provides an overview of the real datasets and the training/validation/testing split sets. We extract 25,071 ICU stays with EHR records from MIMIC-IV, 5,931 of which are matched to CXR images and reports. Similarly, we extract 21,139 ICU stays with EHR records from MIMIC-III, with 5,273 stays matched to clinical notes. To evaluate the performance of CM$^2$ on cross-modal alignment, we conduct experiments on *totally matched* bi-modal and tri-modal settings. We also evaluate *partially matched* datasets to demonstrate the robustness of CM$^2$ in the presence of missing modalities. Further details on the datasets can be found in Appendix A.1.

**Task and Evaluation Metrics.** Following the common practice in clinical prediction tasks (Hayat et al., 2022; Zhang et al., 2022; Wu et al., 2024b; Wang et al., 2024), we focus on two clinical prediction tasks: (1) **In-Hospital Mortality (IHM)** prediction, which predicts whether a patient will pass away during the hospital stay; and (2) **Readmission (READM)** prediction, which aims to predict whether a patient will be readmitted within 30 days after discharge. To assess model performance, we compute the area under the precision-recall curve (AUPR) and the area under the receiver operating characteristic curve (AUROC). Results are reported with the corresponding 95% confidence intervals based on 1,000 bootstrap iterations.

**Backbone Encoders.** Following Hayat et al. (2022), we utilize ResNet34 (He et al., 2016) as the backbone encoder for the CXR image data. For time-series data, we employ a two-layer stacked LSTM network (Graves & Graves, 2012). For clinical notes and radiology reports, we use the TinyBERT encoder (Jiao et al., 2019). A projection layer is applied to map the modality embeddings into the same latent space.

### 4.2. Compared Methods

We compare CM$^2$ against the following baselines: (1) **MMTM** (Joze et al., 2020) is a flexible plugin module that facilitates information exchange between modalities. We address missing CXR and clinical notes during training and testing by filling in the missing data with zeros. (2) **DAFT** (Pölsterl et al., 2021) is a module designed to exchange information between tabular data and image modalities when integrated into CNN models. Similarly, we

*Table 2.* Results of AUROC and AUPR with 95% confidence intervals on MIMIC-III and MIMIC-IV datasets with totally matched modalities. The best results are highlighted in **boldface**.

| Datasets | Models | IHM | | READM | |
|---|---|---|---|---|---|
| | | AUROC (↑) | AUPR (↑) | AUROC (↑) | AUPR (↑) |
| MIMIC-III | MMTM (Joze et al., 2020) | $0.776_{(0.728,0.819)}$ | $0.347_{(0.268,0.447)}$ | $0.716_{(0.670,0.762)}$ | $0.341_{(0.277,0.419)}$ |
| | DAFT (Pölsterl et al., 2021) | $0.792_{(0.746,0.839)}$ | $0.388_{(0.299,0.484)}$ | $0.701_{(0.653,0.746)}$ | $0.325_{(0.262,0.403)}$ |
| | Unified (Hayat et al., 2021) | $0.827_{(0.782,0.868)}$ | $0.466_{(0.371,0.569)}$ | $0.714_{(0.662,0.759)}$ | $0.423_{(0.344,0.504)}$ |
| | MedFuse (Hayat et al., 2022) | $0.826_{(0.781,0.866)}$ | $0.430_{(0.340,0.537)}$ | $0.725_{(0.676,0.774)}$ | $0.414_{(0.338,0.502)}$ |
| | DrFuse (Yao et al., 2024) | $0.835_{(0.793,0.874)}$ | $0.511_{(0.417,0.607)}$ | $0.749_{(0.699,0.795)}$ | $0.441_{(0.356,0.527)}$ |
| | CM$^2$ | $\mathbf{0.854}_{(0.820,0.861)}$ | $\mathbf{0.513}_{(0.460,0.557)}$ | $\mathbf{0.754}_{(0.731,0.774)}$ | $\mathbf{0.445}_{(0.403,0.487)}$ |
| MIMIC-IV | MMTM (Joze et al., 2020) | $0.802_{(0.770,0.835)}$ | $0.429_{(0.362,0.513)}$ | $0.713_{(0.677,0.750)}$ | $0.420_{(0.362,0.489)}$ |
| | DAFT (Pölsterl et al., 2021) | $0.815_{(0.782,0.844)}$ | $0.454_{(0.387,0.538)}$ | $0.729_{(0.692,0.766)}$ | $0.433_{(0.378,0.499)}$ |
| | Unified (Hayat et al., 2021) | $0.808_{(0.778,0.840)}$ | $0.429_{(0.367,0.512)}$ | $0.719_{(0.680,0.756)}$ | $0.450_{(0.390,0.513)}$ |
| | MedFuse (Hayat et al., 2022) | $0.813_{(0.777,0.844)}$ | $0.448_{(0.380,0.528)}$ | $0.725_{(0.690,0.762)}$ | $0.438_{(0.379,0.508)}$ |
| | DrFuse (Yao et al., 2024) | $0.818_{(0.784,0.850)}$ | $0.460_{(0.391,0.540)}$ | $0.726_{(0.689,0.760)}$ | $0.430_{(0.370,0.495)}$ |
| | CM$^2$ | $\mathbf{0.827}_{(0.790,0.859)}$ | $\mathbf{0.492}_{(0.423,0.566)}$ | $\mathbf{0.737}_{(0.704,0.773)}$ | $\mathbf{0.466}_{(0.404,0.529)}$ |

*Table 3.* Results of AUROC and AUPR with 95% confidence intervals on MIMIC-III and MIMIC-IV datasets with partially matched modalities (i.e., missing modalities). The best results are highlighted in **boldface**.

| Datasets | Models | IHM | | READM | |
|---|---|---|---|---|---|
| | | AUROC (↑) | AUPR (↑) | AUROC (↑) | AUPR (↑) |
| MIMIC-III | MMTM (Joze et al., 2020) | $0.846_{(0.825,0.865)}$ | $0.450_{(0.399,0.509)}$ | $0.742_{(0.716,0.766)}$ | $0.413_{(0.371,0.455)}$ |
| | DAFT (Pölsterl et al., 2021) | $0.854_{(0.836,0.873)}$ | $0.495_{(0.440,0.552)}$ | $0.748_{(0.724,0.772)}$ | $0.429_{(0.386,0.473)}$ |
| | Unified (Hayat et al., 2021) | $0.849_{(0.829,0.868)}$ | $0.491_{(0.436,0.542)}$ | $0.751_{(0.728,0.772)}$ | $0.427_{(0.383,0.467)}$ |
| | MedFuse (Hayat et al., 2022) | $0.850_{(0.830,0.868)}$ | $0.480_{(0.426,0.533)}$ | $0.753_{(0.730,0.775)}$ | $0.437_{(0.396,0.480)}$ |
| | DrFuse (Yao et al., 2024) | $0.839_{(0.817,0.861)}$ | $0.474_{(0.422,0.531)}$ | $0.749_{(0.727,0.770)}$ | $0.411_{(0.371,0.455)}$ |
| | CM$^2$ | $\mathbf{0.856}_{(0.833,0.877)}$ | $\mathbf{0.510}_{(0.463,0.566)}$ | $\mathbf{0.754}_{(0.708,0.795)}$ | $\mathbf{0.445}_{(0.358,0.523)}$ |
| MIMIC-IV | MMTM (Joze et al., 2020) | $0.855_{(0.840,0.869)}$ | $0.519_{(0.477,0.561)}$ | $0.765_{(0.747,0.783)}$ | $0.465_{(0.430,0.501)}$ |
| | DAFT (Pölsterl et al., 2021) | $0.857_{(0.841,0.870)}$ | $0.526_{(0.487,0.565)}$ | $0.765_{(0.747,0.782)}$ | $0.476_{(0.442,0.510)}$ |
| | Unified (Hayat et al., 2021) | $0.854_{(0.839,0.870)}$ | $0.505_{(0.463,0.545)}$ | $0.759_{(0.742,0.776)}$ | $0.470_{(0.436,0.503)}$ |
| | MedFuse (Hayat et al., 2022) | $0.855_{(0.840,0.870)}$ | $0.500_{(0.458,0.541)}$ | $0.762_{(0.744,0.778)}$ | $0.465_{(0.430,0.501)}$ |
| | DrFuse (Yao et al., 2024) | $0.857_{(0.841,0.872)}$ | $0.518_{(0.479,0.562)}$ | $0.768_{(0.749,0.784)}$ | $0.485_{(0.451,0.520)}$ |
| | CM$^2$ | $\mathbf{0.858}_{(0.844,0.872)}$ | $\mathbf{0.527}_{(0.490,0.568)}$ | $\mathbf{0.771}_{(0.752,0.788)}$ | $\mathbf{0.486}_{(0.452,0.518)}$ |

replace missing CXR and clinical notes with zero matrices during training and testing. (3) **Unified** (Hayat et al., 2021) is a dynamic approach for integrating auxiliary data modalities and combining all representations via a unified classifier. It handles missing data inherently and leverages all available modality-specific information. (4) **MedFUSE** (Hayat et al., 2022) employs LSTM-based fusion to combine features from image or language encoders with EHR encoders. It handles missing modalities by learning a global representation for absent CXR or clinical notes. (5) **DrFuse** (Yao et al., 2024) leverages disentangled representation learning to create a shared representation between the EHR and image modalities, even when one modality is missing.

*Table 4.* Ablation study on different alignment loss functions with AUROC and AUPR on MIMIC-IV.

| Alignment loss | IHM | | READM | |
|---|---|---|---|---|
| | AUROC (↑) | AUPR (↑) | AUROC (↑) | AUPR (↑) |
| Cosine | 0.820 | 0.470 | 0.726 | 0.445 |
| KL | 0.826 | 0.489 | 0.731 | 0.446 |
| Copula | 0.827 | 0.492 | 0.737 | 0.466 |

### 4.3. Experimental Results

**Quantitative Results.** Table 2 presents results on the MIMIC-III and MIMIC-IV datasets with *totally matched* modalities. CM$^2$ outperforms all the five baselines in all cases. Notably, for the IHM task, CM$^2$ exceeds the best baseline by 1.9% in AUROC on MIMIC-III and 3.2% in AUPR on MIMIC-IV. These results demonstrate the effectiveness

*Table 5.* Ablation study on the influence of different components (e.g., copula alignment, gradient-preserving sampling (GPS), and fusion module) of our proposed method on MIMIC-IV.

| Models | Matched | IHM | | READM | |
|---|---|---|---|---|---|
| | | AUROC (↑) | AUPR (↑) | AUROC (↑) | AUPR (↑) |
| w/o copula | × | 0.855 | 0.506 | 0.753 | 0.459 |
| w/o GPS | × | 0.858 | 0.521 | 0.763 | 0.473 |
| w/o fusion | × | 0.860 | 0.531 | 0.762 | 0.476 |
| $CM^2$ | × | 0.858 | 0.527 | 0.771 | 0.486 |
| w/o copula | ✓ | 0.809 | 0.434 | 0.717 | 0.424 |
| w/o fusion | ✓ | 0.811 | 0.446 | 0.720 | 0.424 |
| $CM^2$ | ✓ | 0.827 | 0.492 | 0.737 | 0.466 |

*Table 6.* Results on different copula families and the influence of the missing modality on MIMIC-IV.

| Matched | Copula | IHM | | READM | |
|---|---|---|---|---|---|
| | | AUROC (↑) | AUPR (↑) | AUROC (↑) | AUPR (↑) |
| × | Gumbel | 0.858 | 0.527 | 0.772 | 0.485 |
| ✓ | Gumbel | 0.825 | 0.488 | 0.735 | 0.463 |
| × | Frank | 0.858 | 0.527 | 0.771 | 0.486 |
| ✓ | Frank | 0.827 | 0.492 | 0.737 | 0.466 |
| × | Gaussian | 0.859 | 0.527 | 0.771 | 0.485 |
| ✓ | Gaussian | 0.827 | 0.488 | 0.736 | 0.458 |

of $CM^2$ in capturing the interactions between modalities and enhancing the performance of multimodal learning tasks in clinical prediction.

Table 3 reports results on the MIMIC-III and MIMIC-IV datasets with *partially matched* modalities (e.g., missing modality). $CM^2$ outperforms the baselines in all cases, with the best performance on the MIMIC-III dataset, where it outperforms the best baseline by 1.5% in AUPR for the IHM task and 0.8% in AUPR for the READM task. This indicates that $CM^2$ effectively learns the joint distribution of the modalities, generating robust and unbiased representations in the presence of missing modalities.

Moreover, our results reveal that the performance on the partially matched datasets is superior to that on the matched datasets. This can be attributed to the larger number of observations in the partially matched datasets, underscoring the importance of multimodal learning in the presence of missing modalities. Lastly, we observe that the performance on MIMIC-IV is better than that on MIMIC-III under the partially matched setting, likely due to the larger number of observations in MIMIC-IV. Additionally, the heterogeneity between modalities in MIMIC-IV may be greater than that in MIMIC-III, contributing to the difference in performance between the two datasets under the totally matched setting.

**Qualitative Analysis.** We visualize the densities of different families of copula and see how the interactions between modalities are captured. Figure 2 presents the visualization of learned densities of the Gumbel, Gaussian, and Frank copula families, respectively. We observe that the Gumbel

copula is more focused on the positive dependence between the modalities while the Gaussian copula has lower weight on modelling tail dependencies. On the other hand, the Frank copula is tail-symmetric and capable of modelling both positive and negative dependencies. Hence, it can cover more dependency structures, indicating that it may be a more flexible choice for modelling complex interactions. We further demonstrate how $CM^2$ learns the interactions through density plots at different epochs. The detailed discussion can be found in Appendix D. We also study how $CM^2$ learns the correlation over epochs. Figure 3 presents the change in the estimated $\alpha$ and its corresponding correlation $\frac{\alpha-1}{\alpha}$ over training epochs. We discover that the model learns a positive correlation over the epochs, and the correlation converges at around 0.601. This implies that by backpropagating the gradient to the copula parameter $\alpha$, the model can learn the interactions between the modalities during training.

### 4.4. Ablation Analysis

**Effectiveness of Copula Alignment.** We study the effects of the alignment loss, as presented in Table 4. The copula alignment loss achieves the best performance, outperforming the popular cosine similarity alignment and KL divergence alignment.

**Ablation on Contribution of the Designed Modules.** To further evaluate the performance of $CM^2$, we conduct an ablation study by removing the copula alignment, the gradient-preserving sampling (GPS), and fusion modules, respectively. As shown in Table 5, the performance of $CM^2$ significantly declines without copula alignment, underscoring the importance of modeling the copula joint distribution before fusing modality features. Additionally, in most cases, removing the fusion module leads to a notable drop in performance, emphasizing its critical role in capturing modality interactions. Furthermore, we observe a slight decline when the GPS is removed, indicating its effectiveness in generating unbiased representations for observations with missing modalities.

**Ablation on Different Families of Copula.** We also compare the performance of $CM^2$ under different settings for missing modalities and copula families. The accuracy relies heavily on the assumed copula family (Zeng & Wang, 2022). We examine the performance of our method over an array of commonly used copula families. Table 6 presents the results of $CM^2$ on the MIMIC-IV dataset. We discover that while our method is generally robust to the choice of copula family, the best-performing copula varies across different tasks. This indicates that different tasks highlight different characteristics (e.g., extreme values for mortality) that can be captured when a proper copula family is chosen.

**Extension to More Modalities.** We further investigate the

*Table 7.* Results of AUROC and AUPR with 95% confidence intervals using three modalities (EHR time series, CXR images, and CXR reports) on MIMIC-IV.

| Models | IHM | | READM | |
|---|---|---|---|---|
| | AUROC (↑) | AUPR (↑) | AUROC (↑) | AUPR (↑) |
| MMTM (Joze et al., 2020) | $0.777_{(0.739,0.813)}$ | $0.370_{(0.312,0.443)}$ | $0.689_{(0.650,0.723)}$ | $0.401_{(0.347,0.463)}$ |
| DAFT (Pölsterl et al., 2021) | $0.788_{(0.754,0.821)}$ | $0.397_{(0.331,0.471)}$ | $0.706_{(0.670,0.742)}$ | $0.403_{(0.346,0.464)}$ |
| Unified (Hayat et al., 2021) | $0.795_{(0.761,0.827)}$ | $0.420_{(0.351,0.497)}$ | $0.715_{(0.679,0.749)}$ | $0.430_{(0.376,0.495)}$ |
| MedFuse (Hayat et al., 2022) | $0.801_{(0.767,0.836)}$ | $0.427_{(0.367,0.511)}$ | $0.713_{(0.675,0.749)}$ | $0.419_{(0.356,0.487)}$ |
| DrFuse (Yao et al., 2024) | $0.808_{(0.773,0.839)}$ | $0.451_{(0.376,0.524)}$ | $0.728_{(0.691,0.761)}$ | $0.433_{(0.370,0.495)}$ |
| $\text{CM}^2$ | $\mathbf{0.824}_{(0.793,0.856)}$ | $\mathbf{0.471}_{(0.399,0.554)}$ | $\mathbf{0.730}_{(0.694,0.764)}$ | $\mathbf{0.444}_{(0.385,0.509)}$ |

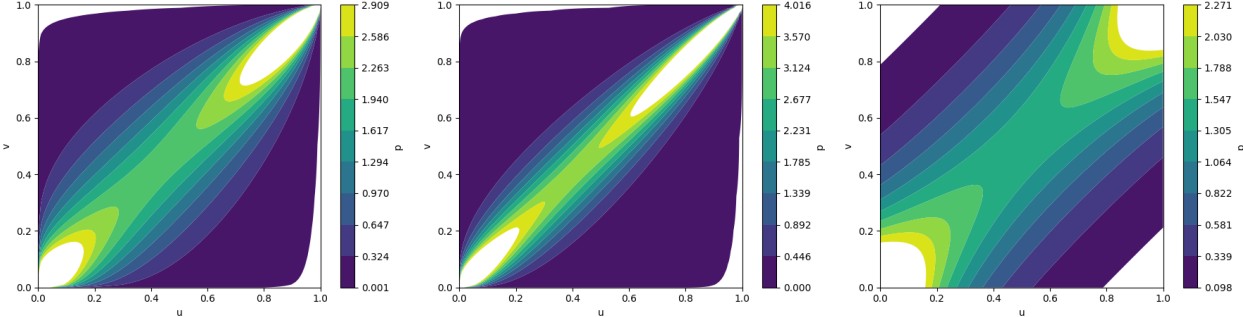

*Figure 2.* Plots of the fitted copula density to demonstrate the interrelationship captured by the copula model (Left: Gumbel, middle: Gaussian, right: Frank).

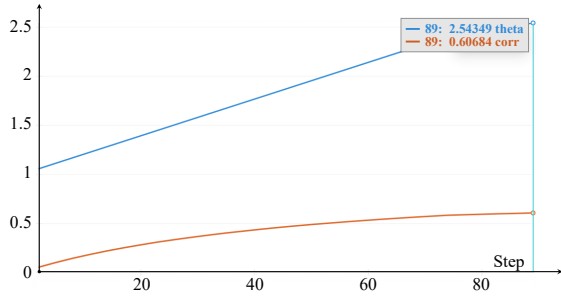

*Figure 3.* Plots comparing the value of $\alpha$ and the correlation, $\text{Corr} = (\alpha - 1)/\alpha$ learned by the Gumbel copula model.

impact of incorporating more auxiliary modalities. We adapt all baselines into the tri-modal setting. Table 7 presents the results for $\text{CM}^2$ and the baselines on the MIMIC-IV dataset under the tri-modal setting: EHR time series, CXR images, and radiology reports. Across both tasks, $\text{CM}^2$ consistently outperforms the baselines, achieving the best performance. Notably, the baseline models show a decline in performance compared to the bi-modal setting, suggesting that incorporating additional modalities becomes more challenging as the alignment complexity increases. Despite this, $\text{CM}^2$ maintains strong performance, demonstrating its robustness and effectiveness in aligning multiple modalities.

## 5. Conclusion

We introduce copula modelling into multimodal representation learning. Using a copula can effectively model the interactions among different modalities, and impute the missing modalities through sampling from learned marginals. Empirical evaluation validates the predictive performance on the multimodal learning tasks, on both the fully and partially matched datasets. Ablation studies show that the proposed copula model can serve as a promising modality alignment tool due to the consistently satisfactory performance over different copula families. Our idea can be potentially extended to works that require effective fusion or distribution alignment, including domain adaptation, multi-feature and multi-view learning.

**Limitations and Future Works.** Using a neural network to learn the copula parameter $\alpha$ may be insufficient (since the joint log-likelihood may not be convex). Hence, an alternative updating algorithm (e.g., partial likelihood) is needed in future development of copula multimodal learning to ensure that each loss is convex to apply gradient descent. While we select healthcare datasets to demonstrate the effectiveness of our model, our method can be extended to other types of multimodal datasets.

**Acknowledgement.** We thank the program chairs, area chairs, and reviewers for many constructive suggestions

that have significantly improved the paper. This work was supported in part by the Research Grants Council of Hong Kong (17308321, 27206123, C5055-24G, and T45-401/22-N), Patrick SC Poon endowment fund, Hong Kong Innovation and Technology Fund (ITS/273/22 and ITS/274/22), National Natural Science Foundation of China (No. 62201483), and Guangdong Natural Science Fund (No. 2024A1515011875).

## Impact Statement

The goal of this work is to advance the field of machine learning. Our work can be potentially extended to areas that require effective fusion or distribution alignment, including domain adaptation, multi-feature learning, and multi-view learning.

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

## Summary

In this Appendix, we first present detailed information on the datasets in A.1 and tasks used in the experiments in A.2. Next, we introduce the multivariate Gaussian distribution in B and some common copula families in C. Then in D, we discuss the implications of how the copula model learns interactions over the epochs. Finally, we provide more details on the implementation and hyperparameters used in the experiments in E.1 along with the settings of baseline methods in E.2.

## A. Additional Information on Datasets and Tasks

### A.1. Datasets

Table 8 provides a summary of the datasets used in our experiments.

**MIMIC-III dataset** This dataset contains 46,520 ICU stays, each with 17 clinical variables. We split the dataset into training, validation, and test sets in the ratio of 70 : 15 : 15, following the procedure in Harutyunyan et al. (2019).

**MIMIC-IV dataset** This dataset includes 21,139 ICU stays, also with 17 clinical variables. The dataset is split into training, validation, and test sets in the ratio of 70 : 10 : 20, following Hayat et al. (2022).

For both MIMIC-III and MIMIC-IV datasets, we extract 17 clinical variables commonly monitored in the ICU, including 5 categorical and 12 continuous variables. Data are sampled every two hours during the first 48 hours of ICU admission for both tasks, in accordance with Hayat et al. (2022). This results in a vector representation of size 76 at each time step of the clinical time-series data.

**MIMIC-CXR dataset** This dataset contains 377,110 chest X-ray images, of which 5,931 are associated with MIMIC-IV ICU stays. We split the data into 4,287 training samples, 465 validation samples, and 1,179 test samples. Following Hayat et al. (2022), we retrieve the last Anterior-Posterior projection chest X-ray and apply transformations to the images, resizing them to $224 \times 224$ pixels.

This dataset also includes radiology reports, which are unstructured text data. We choose the radiology reports of the MIMIC-CXR dataset as an auxiliary modality to investigate the effectiveness of $CM^2$ on more modalities alignment since the radiology reports do not contain death information and can avoid possible overfitting and shortcuts. We divide the unstructured radiology reports into 4 sections, including Impression, Findings, Last paragraph, and Comparison.

**MIMIC-III NOTE dataset** This dataset consists of 5,273 clinical notes associated with MIMIC-III ICU stays. The dataset is divided into 3,652 training samples, 815 validation samples, and 806 test samples. In line with Zhang et al. (2023), we select the last five clinical notes before the prediction time. If fewer than five notes are available, we treat the notes for that ICU stay as missing. The original number of matched ICU stays is around 15,000. We randomly sample one-third of the matched ICU stays to form the training, validation, and test sets, keeping the scale of the notes nearly the same as the CXRs in the MIMIC-IV dataset.

Both radiology reports sections and clinical notes are capped at a maximum length of 512 words, tokenized into words, and embedded into 312-dimensional vectors using the pre-trained TinyBERT model (Jiao et al., 2019)[1].

### A.2. Tasks

**In-Hospital Mortality (IHM) Prediction.** The In-Hospital Mortality (IHM) prediction task focuses on predicting whether a patient will pass away during their hospital stay. As summarized in Table 8, the MIMIC-III dataset contains a total of 2,795 positive samples, of which 736 are matched with clinical notes. Similarly, the MIMIC-IV dataset includes 3,153 positive samples, with 890 matched to CXR.

**Readmission (READM) Prediction.** The Readmission (READM) prediction task aims to forecast whether a patient will be readmitted within 30 days of discharge. In this task, both patients who are readmitted and those who pass away in hospital are considered positive samples. As shown in Table 8, the MIMIC-III dataset contains 3,987 positive samples, with 998 matched to clinical notes. In the MIMIC-IV dataset, there are 4,603 positive samples, with 1,262 matched to CXRs.

---

[1] https://huggingface.co/huawei-noah/TinyBERT_General_4L_312D

*Table 8.* Numbers of samples in training/validation/testing sets

| Datasets | Tasks | Train | Valid | Test | Pos. | Total |
|---|---|---|---|---|---|---|
| | | *Complete Datasets* | | | | |
| **MIMIC-III** | IHM | 14681 | 3222 | 3236 | 2795 | 21139 |
| **MIMIC-III** | READM | 14681 | 3222 | 3236 | 3987 | 21139 |
| **MIMIC-III NOTE** | – | 3652 | 815 | 806 | – | 5,273 |
| **MIMIC-IV** | IHM | 18064 | 2035 | 4972 | 3153 | 25071 |
| **MIMIC-IV** | READM | 18064 | 2035 | 4972 | 4603 | 25071 |
| **MIMIC-CXR** | – | 344529 | 9497 | 23069 | – | 377,095 |
| | | *Matched Datasets* | | | | |
| **MIMIC-III \| NOTE** | IHM | 3652 | 815 | 806 | 736 | 5273 |
| **MIMIC-III \| NOTE** | READM | 3652 | 815 | 806 | 998 | 5273 |
| **MIMIC-IV \| CXR** | IHM | 4287 | 465 | 1179 | 890 | 5931 |
| **MIMIC-IV \| CXR** | READM | 4287 | 465 | 1179 | 1262 | 5931 |
| **MIMIC-IV \| CXR \| REPORT** | IHM | 4287 | 465 | 1179 | 890 | 5931 |
| **MIMIC-IV \| CXR \| REPORT** | READM | 4287 | 465 | 1179 | 1262 | 5931 |

# B. Multivariate Gaussian Distribution

The multivariate Gaussian distribution is defined as

$$p(\boldsymbol{z}; \boldsymbol{\mu}, \boldsymbol{\Sigma}) = \frac{1}{(2\pi)^{\frac{n}{2}} |\boldsymbol{\Sigma}|^{\frac{1}{2}}} \exp\left\{ -\frac{1}{2}(\boldsymbol{z} - \boldsymbol{\mu})^\top \boldsymbol{\Sigma}^{-1}(\boldsymbol{z} - \boldsymbol{\mu}) \right\},$$

where $\boldsymbol{\mu} \in \mathbb{R}^p$ is a $p$-dimensional mean vector and $\boldsymbol{\Sigma} \in \mathbb{R}^{p \times p}$ is the covariance matrix.

The KL divergence of two multivariate normal distributions $\mathcal{N}(\boldsymbol{\mu}_1, \Sigma_1)$ and $\mathcal{N}(\boldsymbol{\mu}_2, \Sigma_2)$ is

$$\mathrm{KL}(\mathcal{N}(\boldsymbol{\mu}_1, \boldsymbol{\Sigma}_1) \| \mathcal{N}(\boldsymbol{\mu}_2, \boldsymbol{\Sigma}_2)) = \frac{1}{2}\left[ \log\frac{|\boldsymbol{\Sigma}_2|}{|\boldsymbol{\Sigma}_1|} - p + \mathrm{tr}\{\boldsymbol{\Sigma}_2^{-1}\boldsymbol{\Sigma}_1\} + (\boldsymbol{\mu}_2 - \boldsymbol{\mu}_1)^\top \boldsymbol{\Sigma}_2^{-1}(\boldsymbol{\mu}_2 - \boldsymbol{\mu}_1) \right].$$

# C. Common Copula Families.

We specify the copula distributions and density functions of common copula families with necessary derivations. Without loss of generality, we consider bivariate copula families.

**Archimedean Copula.** A subclass of copulas can be easily constructed by a generator function $\varphi : [0, 1] \to [0, \infty]$, which is strictly decreasing and convex so that $\varphi(0) = \infty$ and $\varphi(1) = 0$. Then, a copula $C$ can be constructed as follows,

$$C(u_1, u_2, \ldots, u_d) = \varphi^{[-1]}\left( \sum_{i=1}^{d} \varphi(u_i) \right).$$

The Archimedean copula can generate copula densities when more than one modality exist in the dataset.

### C.1. Copula Distribution Functions

- Clayton copula

$$C(u, v; \alpha) = \left[ \max\{u^{-\alpha} + v^{-\alpha} - 1, 0\} \right]^{-1/\alpha}.$$

- Frank copula

$$C(u, v; \alpha) = -\frac{1}{\alpha} \log\left[ 1 - \frac{(1 - e^{\alpha u})(1 - e^{\alpha v}))}{1 - e^{-\alpha}} \right],$$

where $\alpha \in \mathbb{R}\backslash\{0\}$.

- Gumbel copula

$$C(u, v; \alpha) = \exp\{-[(-\log u)^\alpha + (-\log v)^\alpha]^{\frac{1}{\alpha}}\}.$$

- Gaussian copula

$$C(u, v; \rho) = \Phi_2\left[\Phi^{-1}(u), \Phi^{-1}(v); \rho\right],$$

where $\Phi$ is the CDF of the standard Gaussian distribution, and $\Phi_2$ is the bivariate Gaussian distribution.

- Student's $t$ copula

$$C(u, v; \rho, \nu) = T_{2,\nu}[T_\nu^{-1}(u), T_\nu^{-1}(v); \rho], \qquad v > 0; |\rho| < 1,$$

where $T_\nu^{-1}$ is the inverse of the CDF of Student's $t$-distribtuion with degrees of freedom $\nu$, and $T_{2,\nu}$ is the bivariate $t$-distribtuion with degrees of freedom $\nu$.

## C.2. Copula Density Functions

**Clayton copula**

$$c(u, v) = (1 + \alpha)(uv)^{-1-\alpha}(-1 + u^{-\alpha} + v^{-\alpha})^{-2-1/\alpha},$$

where $\alpha \in (-1, \infty)$.

**Frank copula**

$$c(u, v) = \frac{-\alpha e^{-\alpha(u+v)}(e^{-\alpha} - 1)}{(e^{-\alpha} - e^{-\alpha u} - e^{-\alpha v} + e^{-\alpha(u+v)})^2},$$

where $\alpha \in (-\infty, \infty), \alpha \neq 0$.

**Gumbel copula**

$$
\begin{aligned}
c(u,v) =& \frac{\partial}{\partial u}\frac{\partial}{\partial v}C(u,v) \\
=& \frac{\partial}{\partial u}\frac{\partial}{\partial v}\exp\{-[(-\log u)^\alpha + (-\log v)^\alpha]^{1/\alpha}\} \\
:=& \frac{\partial}{\partial u}\frac{\partial}{\partial v}\exp\{-[g(u,v;\alpha)]^{1/\alpha}\} \\
=& \frac{\partial}{\partial u} - \exp\{-[g(u,v;\alpha)]^{1/\alpha}\}\left(\frac{1}{\alpha}[g(u,v;\alpha)]^{\frac{1-\alpha}{\alpha}}\right)\frac{\partial}{\partial v}g(u,v;\alpha) \\
=& \frac{\partial}{\partial u}\exp\{-[g(u,v;\alpha)]^{1/\alpha}\}\left(\frac{1}{\alpha}[g(u,v;\alpha)]^{\frac{1-\alpha}{\alpha}}\right)\alpha(-\log v)^{\alpha-1}\frac{1}{v} \\
=& \frac{\alpha}{v}(-\log v)^{\alpha-1}\left[\left(\frac{1}{\alpha}[g(u,v;\alpha)]^{\frac{1-\alpha}{\alpha}}\right)\frac{\partial}{\partial u}\exp\{-[g(u,v;\alpha)]^{1/\alpha}\}\right. \\
& \left. + \exp\{-[g(u,v;\alpha)]^{1/\alpha}\}\frac{\partial}{\partial u}\frac{1}{\alpha}[g(u,v;\alpha)]^{\frac{1-\alpha}{\alpha}}\right] \\
=& \frac{\alpha}{v}(-\log v)^{\alpha-1}\left[-\frac{1}{\alpha}[g(u,v;\alpha)]^{\frac{1-\alpha}{\alpha}}\exp\{-[g(u,v;\alpha)]^{1/\alpha}\}\left(\frac{1}{\alpha}[g(u,v;\alpha)]^{\frac{1-\alpha}{\alpha}}\right)\frac{\partial}{\partial u}g(u,v;\alpha)\right. \\
& \left. + \exp\{-[g(u,v;\alpha)]^{1/\alpha}\}\frac{1-\alpha}{\alpha^2}[g(u,v;\alpha)]^{\frac{1-2\alpha}{\alpha}}\frac{\partial}{\partial u}g(u,v;\alpha)\right] \\
=& \frac{\alpha}{v}(-\log v)^{\alpha-1}\frac{\partial}{\partial u}g(u,v;\alpha)\exp\{-[g(u,v;\alpha)]^{1/\alpha}\}\left[\frac{1}{\alpha^2}[g(u,v;\alpha)]^{\frac{2(1-\alpha)}{\alpha}}\right. \\
& \left. + \frac{\alpha-1}{\alpha^2}[g(u,v;\alpha)]^{\frac{1-2\alpha}{\alpha}}\right] \\
=& \frac{1}{uv}(-\log v)^{\alpha-1}(-\log u)^{\alpha-1}C(u,v)\left[(\alpha-1)[g(u,v;\alpha)]^{\frac{1-2\alpha}{\alpha}} + [g(u,v;\alpha)]^{\frac{2(1-\alpha)}{\alpha}}\right] \\
=& \frac{1}{uv}(-\log v)^{\alpha-1}(-\log u)^{\alpha-1}C(u,v)[g(u,v;\alpha)]^{\frac{2(1-\alpha)}{\alpha}}\left[(\alpha-1)[g(u,v;\alpha)]^{-\frac{1}{\alpha}} + 1\right].
\end{aligned}
$$

The closed-form density of the trivariate Gumbel copula is computed by

$$
\begin{aligned}
c(u,v,w) =& \frac{\partial}{\partial u}\frac{\partial}{\partial v}\frac{\partial}{\partial w}C(u,v,w) \\
=& \frac{\partial}{\partial u}\frac{\partial}{\partial v}\frac{\partial}{\partial w}\exp\{-[(-\log u)^\alpha + (-\log v)^\alpha + (-\log w)^\alpha]^{1/\alpha}\} \\
:=& \frac{\partial}{\partial u}\frac{\partial}{\partial v}\frac{\partial}{\partial w}\exp\{-(h(u,v,w;\alpha))^\alpha\} \\
=& \frac{1}{uvw}(-\log(u))^{\alpha-1}(-\log(v))^{\alpha-1}(-\log(w))^{\alpha-1}C(u,v,w) \\
& \cdot \left(\alpha^6\,(h(u,v,w;\alpha))^{3\alpha-3} - (\alpha-1)\alpha^5\,(h(u,v,w;\alpha))^{2\alpha-3} - 2\alpha^5(\alpha-1)\,(h(u,v,w;\alpha))^{2\alpha-3}\right. \\
& \left. + (\alpha-2)(\alpha-1)\alpha^4\,(h(u,v,w;\alpha))^{\alpha-3}\right).
\end{aligned}
$$

The identity can be generated by the Archimedean copula for $M > 3$, which is less common in multimodal learning,

$$
c(\boldsymbol{u}) = \varphi^{(d)}(t(\boldsymbol{u}))\prod_{j=1}^{d}(\varphi^{-1})'(u_j),
$$

where $\varphi(t;\alpha) = (\log t)^\alpha$ for the Gumbel copula, and $(\varphi^{-1})'$ is the first derivative of the inverse of $\varphi$.

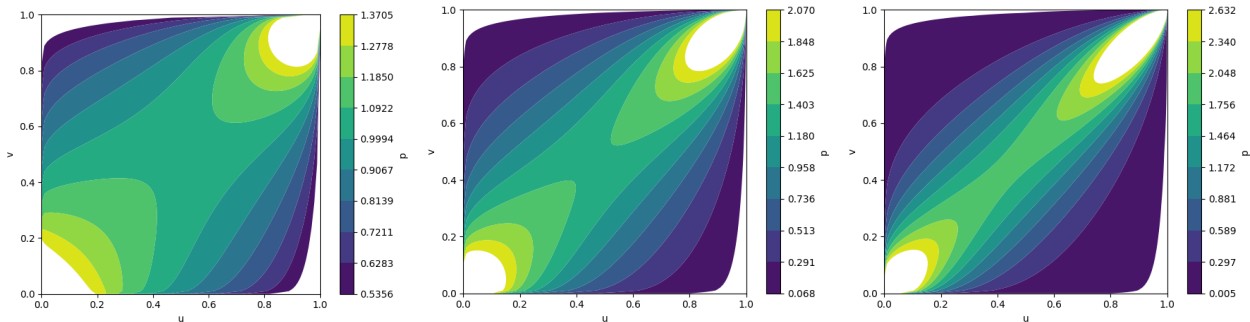

*Figure 4.* Plots of the copula densities of the Gumbel family at epochs 5, 50, and 100, respectively.

**Gaussian copula** The bivariate case is given by

$$c(u, v; \rho) = \frac{1}{\sqrt{1 - \rho}} \exp\left(-\frac{(a^2 + b^2)\rho^2 - 2ab\rho}{2(1 - \rho^2)}\right),$$

where $a = \sqrt{2}\text{erf}^{-1}(2u - 1)$, and $b = \sqrt{2}\text{erf}^{-1}(2v - 1)$. The multivariate case is given by the following matrix form,

$$c(\boldsymbol{u}; \boldsymbol{\Sigma}) = (2\pi)^{M/2}|\boldsymbol{\Sigma}|^{-1/2}\frac{\exp\{-\zeta^\top\boldsymbol{\Sigma}^{-1}\zeta/2\}}{\prod_{m=1}^{M}(2\pi)^{-1/2}\exp(-\zeta_m^2/2)},$$

where $\boldsymbol{\Sigma}$ is the covariance matrix and $\zeta = (\Phi^{-1}(u_1), \ldots, \Phi^{-1}(u_M))^\top$, $\zeta = [\zeta_1, \ldots, \zeta_M]$.

**Student's $t$ copula**

$$\frac{\Gamma(v/2)\Gamma(v/2 + 1)(1 + (t_v^{-2}(u) + t_v^{-2}(v) - 2\rho t_v^{-1}(u)t_v^{-1}(v))/(v(1 - \rho^2))^{-(v+2)/2})}{\sqrt{1 - \rho^2}\Gamma((v + 1)/2)^2(1 + t_v^{-2}(u)/v)^{-(v+1)/2}(1 + t_v^{-2}(v)/v)^{-(v+1)/2}},$$

where $v$ is the degree of freedom, $\Gamma$ is the gamma function, and

$$t_v(x) = \int_{-\infty}^{x} \frac{\Gamma((v + 1)/2)dt}{\sqrt{v\pi}\Gamma(v/2)(1 + v^{-1}t^2)^{(v+1)/2}}.$$

## D. How Copula Learns Interactions.

We demonstrate how the copula model learns the interactions over the epochs and further discuss the implications.

Figure 4 presents the copula densities at epochs epochs 5, 50, and 100, respectively. We use the Gumbel family as an illustrative example. We observe that the copula density is evolving to a positive correlation pattern, while the negative correlation scenarios (e.g., $u > 0.5, v < 0.5$, or $u < 0.5, v > 0.5$) are still considered but the weights allocated are decreasing.

## E. More on Baseline Methods and Implementation Details

### E.1. Implementation Details and Hyperparameters

We train all models for 100 epochs on the training set and select the best-performing model based on the validation set, using the AUROC as the monitoring metric. The final results are reported on the test set. We optimize the models using the *Adam* optimizer and apply early stopping if the validation AUROC does not improve for 15 consecutive epochs to prevent overfitting. All experiments are conducted on a single RTX-3090 GPU. The batch size is set to 32 for models trained on the MIMIC-IV & CXR datasets, and 16 for models trained on the MIMIC-III & NOTE datasets, except for DrFuse, which is trained with a batch size of 8. We employ grid search to tune hyperparameters using the validation set and report the best results on the test set. The hyperparameter search space includes:

- Dropout ratio: $\{0, 0.1, 0.2, 0.3\}$

- Learning rate: $\{1 \times 10^{-4}, 5 \times 10^{-5}, 1 \times 10^{-5}\}$

- Number of Gaussian mixtures $K$: $\{1, 2, 3, 4, 5, 6\}$

- Temperature: $\{0.001, 0.005, 0.01, 0.05, 0.08\}$

- Regularization parameter $\lambda_{\text{cop}}$: $\{1 \times 10^{-5}, 5 \times 10^{-6}, 1 \times 10^{-6}\}$

$\text{CM}^2$ is implemented in Python 3.11 using *PyTorch* 1.9. Following MedFuse (Hayat et al., 2022), we use ResNet34 (He et al., 2016) as the backbone encoder for CXR, a two-layer LSTM (Graves & Graves, 2012) as the encoder for time-series data, and pre-trained TinyBERT (Jiao et al., 2019)[2] as the encoder for clinical notes. We include a projection layer to map modality embeddings into the same latent space. A two-layer LSTM is used as the fusion module to combine modality embeddings, and a multilayer perceptron (MLP) with one linear layer and a sigmoid activation function serves as the classifier.

### E.2. Additional Settings of Baseline Methods

We compare $\text{CM}^2$ with the following baseline methods.

- **MMTM** (Joze et al., 2020) is a module that can leverage the information between modalities with flexible plugin architectures. Since the model assumes full modality, we compensate for the missing modality CXR and clinical notes with all zeros during training and testing. For clinical notes, we replace the ResNet34 encoder with TinyBERT to embed the clinical notes.

- **DAFT** (Pölsterl et al., 2021) is a module that can be plugged into CNN models to exchange information between tabular data and image modality. Similarly, we replace the input of CXR and clinical notes with matrices of all zeros during training and testing and use TinyBERT to embed the clinical notes.

- **Unified** (Hayat et al., 2021) is a dynamic approach towards integrating auxiliary data modalities, learning the data representations for the individual modalities, and integrating the representations via a unified classifier. It inherently handles missingness and leverages all of the available modality-specific data. Also, we use TinyBERT to embed the clinical notes.

- **MedFuse** (Hayat et al., 2022) uses an LSTM-based fusion to combine features from the image encoder (or language encoder) and EHR encoder. Missing modality is handled by learning a global representation for the missing CXR or clinical notes. We randomly initialized encoders for the time-series data, clinical notes, and CXR images.

- **DrFuse** (Yao et al., 2024) uses disentangled representation learning to learn a shared representation between the EHR and image modality even when one modality is missing. Drfuse uses ResNet50 as the image encoder and Transformer as the EHR encoder. We replace the ResNet50 encoder with TinyBERT to embed the clinical notes.

The Implementation of DrFuse follows the original paper(Yao et al., 2024)[3], and we use the same hyperparameters as the original paper. We directly adopt the implementations of MMTM, DAFT, Unified, and MedFuse provided by (Hayat et al., 2022)[4], and all hyperparameters are set to the default values provided by Hayat et al. (2022). We adapt the implementations of MMTM, DAFT, Unified, MedFuse and DrFuse to tri-modal setting, including EHR time-series data, CXR images, and radiology reports.

## F. Additional Experiment Results

**Additional Baselines.** We compare $\text{CM}^2$ to two additional healthcare baselines: LSMT (Khader et al., 2023) and Interleaved (Zhang et al., 2023). The results are shown in Table 9.

---

[2] https://huggingface.co/huawei-noah/TinyBERT_General_4L_312D
[3] https://github.com/dorothy-yao/drfuse
[4] https://github.com/nyuad-cai/MedFuse

*Table 9.* Results of additional baselines on the MIMIC-IV dataset. All results are reported in AUROC and AUPR with 95% confidence intervals. The best results are highlighted in **boldface.**

| Models | IHM | | READM | |
|---|---|---|---|---|
| | AUROC ($\uparrow$) | AUPR ($\uparrow$) | AUROC ($\uparrow$) | AUPR ($\uparrow$) |
| | Totally Matched | | | |
| LSMT (Khader et al., 2023) | $0.803_{(0.769,0.837)}$ | $0.444_{(0.370,0.519)}$ | $0.701_{(0.662,0.737)}$ | $0.421_{(0.356,0.490)}$ |
| Interleaved (Zhang et al., 2023) | $0.800_{(0.764,0.834)}$ | $0.440_{(0.374,0.523)}$ | $0.702_{(0.664,0.741)}$ | $0.421_{(0.360,0.487)}$ |
| CM$^2$ | $\mathbf{0.827}_{(0.790,0.859)}$ | $\mathbf{0.492}_{(0.423,0.566)}$ | $\mathbf{0.737}_{(0.704,0.773)}$ | $\mathbf{0.466}_{(0.404,0.529)}$ |
| | Partially Matched | | | |
| LSMT (Khader et al., 2023) | $0.854_{(0.838,0.870)}$ | $0.508_{(0.466,0.551)}$ | $0.764_{(0.746,0.781)}$ | $0.473_{(0.436,0.509)}$ |
| Interleaved (Zhang et al., 2023) | $0.856_{(0.840,0.871)}$ | $0.508_{(0.466,0.550)}$ | $0.758_{(0.740,0.775)}$ | $0.473_{(0.441,0.506)}$ |
| CM$^2$ | $\mathbf{0.858}_{(0.844,0.872)}$ | $\mathbf{0.527}_{(0.490,0.568)}$ | $\mathbf{0.771}_{(0.752,0.788)}$ | $\mathbf{0.486}_{(0.452,0.518)}$ |

*Figure 5.* Results (left: AUROC; right: AUPR) of CM$^2$ on MIMIC-IV, where the model reduces to a multivariate Gaussian disdtribution when $K = 1$.

- **LSMT** (Khader et al., 2023) is a transformer-based model designed for the multimodal medical context.

- **Interleaved** (Zhang et al., 2023) is a multimodal approach that addresses the irregularity of medical multimodal data and fuses representations from different modalities using cross-modal attention.

**Effect of Backbone Encoders.** Moreover, we explore the effectiveness of backbone encoders for both time-series data and CXR image data. We conduct additional experiments to evaluate the impact of different encoder architectures for each modality. Specifically, we use the Transformer (Vaswani, 2017) and ViT (Dosovitskiy, 2020) as alternative backbone encoders for the time-series and CXR image data, respectively. The results are shown in Table 10. We observe that our method consistently outperforms competitive baselines across various backbone encoders, highlighting its robustness and effectiveness. Furthermore, our method demonstrates greater stability across different backbones, suggesting it is less sensitive to their selection. Besides, the Transformer backbone generally outperforms the LSTM backbone, particularly for MMTM, LSMT, and Interleaved. While the ResNet backbone slightly outperforms the ViT backbone, the performance difference is not substantial, suggesting time-series data's greater impact on backbone encoder choice.

**Effect of Number of Mixtures $K$.** As a convention in statistical modelling, $K$ is set to be small to avoid over-specification. The popular choice of $K$ is 2 to 3 such that the learned mixture distribution can achieve an optimal degree of flexibility while preventing over-specification. We evaluate how the performance of CM$^2$ varies with different values of $K$, as shown in Figure 5. We observe that the performance is quite robust.

**Statistical Tests** The p-values of two-sample bootstrapped $t$-tests of the AUROC and AUPR of CM$^2$ compared to baseline methods are shown in Table 11. We observe that the improvements over the competitive baselines are overall statistically significant under the 5% significance level, validating the effectiveness of our method.

*Table 10.* Results of different backbone encoders and additional baselines on MIMIC-IV with totally matched modalities. All results are reported in AUROC and AUPR with 95% confidence intervals. The best results are highlighted in **boldface**.

| Models | Backbone | | IHM | | READM | |
|---|---|---|---|---|---|---|
| | TS | IMG | AUROC (↑) | AUPR (↑) | AUROC (↑) | AUPR (↑) |
| MMTM (Joze et al., 2020) | | | $0.802_{(0.770, 0.835)}$ | $0.429_{(0.362, 0.513)}$ | $0.713_{(0.677, 0.750)}$ | $0.420_{(0.362, 0.489)}$ |
| DAFT (Pölsterl et al., 2021) | | | $0.815_{(0.782, 0.844)}$ | $0.454_{(0.387, 0.538)}$ | $0.729_{(0.692, 0.766)}$ | $0.433_{(0.378, 0.499)}$ |
| Unified (Hayat et al., 2021) | LSTM | ResNet | $0.808_{(0.778, 0.840)}$ | $0.429_{(0.367, 0.512)}$ | $0.719_{(0.680, 0.756)}$ | $0.450_{(0.390, 0.513)}$ |
| MedFuse (Hayat et al., 2022) | | | $0.813_{(0.777, 0.844)}$ | $0.448_{(0.380, 0.528)}$ | $0.725_{(0.690, 0.762)}$ | $0.438_{(0.379, 0.508)}$ |
| DrFuse (Yao et al., 2024) | | | $0.814_{(0.780, 0.844)}$ | $0.450_{(0.384, 0.536)}$ | $0.723_{(0.687, 0.756)}$ | $0.422_{(0.367, 0.486)}$ |
| LSMT (Khader et al., 2023) | | | $0.803_{(0.769, 0.837)}$ | $0.444_{(0.374, 0.523)}$ | $0.701_{(0.662, 0.737)}$ | $0.421_{(0.356, 0.490)}$ |
| Interleaved (Zhang et al., 2023) | | | $0.800_{(0.764, 0.834)}$ | $0.440_{(0.370, 0.519)}$ | $0.702_{(0.664, 0.741)}$ | $0.421_{(0.360, 0.487)}$ |
| CM$^2$ | | | $\mathbf{0.827}_{(0.790, 0.859)}$ | $\mathbf{0.492}_{(0.423, 0.566)}$ | $\mathbf{0.737}_{(0.704, 0.773)}$ | $\mathbf{0.466}_{(0.404, 0.529)}$ |
| MMTM (Joze et al., 2020) | | | $0.805_{(0.768, 0.837)}$ | $0.446_{(0.377, 0.524)}$ | $0.712_{(0.676, 0.749)}$ | $0.422_{(0.360, 0.491)}$ |
| DAFT (Pölsterl et al., 2021) | | | $0.808_{(0.775, 0.840)}$ | $0.438_{(0.365, 0.521)}$ | $0.714_{(0.678, 0.753)}$ | $0.423_{(0.369, 0.490)}$ |
| Unified (Hayat et al., 2021) | LSTM | ViT | $0.803_{(0.768, 0.835)}$ | $0.431_{(0.365, 0.515)}$ | $0.707_{(0.667, 0.743)}$ | $0.416_{(0.360, 0.482)}$ |
| MedFuse (Hayat et al., 2022) | | | $0.805_{(0.771, 0.837)}$ | $0.439_{(0.371, 0.524)}$ | $0.715_{(0.677, 0.753)}$ | $0.424_{(0.370, 0.492)}$ |
| DrFuse (Yao et al., 2024) | | | $0.806_{(0.772, 0.838)}$ | $0.446_{(0.379, 0.526)}$ | $0.716_{(0.677, 0.748)}$ | $0.421_{(0.364, 0.489)}$ |
| LSMT (Khader et al., 2023) | | | $0.801_{(0.767, 0.836)}$ | $0.441_{(0.374, 0.527)}$ | $0.703_{(0.662, 0.739)}$ | $0.410_{(0.358, 0.475)}$ |
| Interleaved (Zhang et al., 2023) | | | $0.802_{(0.766, 0.833)}$ | $0.434_{(0.364, 0.509)}$ | $0.710_{(0.673, 0.747)}$ | $0.435_{(0.372, 0.502)}$ |
| CM$^2$ | | | $\mathbf{0.826}_{(0.790, 0.856)}$ | $\mathbf{0.490}_{(0.421, 0.563)}$ | $\mathbf{0.736}_{(0.697, 0.771)}$ | $\mathbf{0.452}_{(0.394, 0.522)}$ |
| MMTM (Joze et al., 2020) | | | $0.813_{(0.780, 0.846)}$ | $0.452_{(0.383, 0.540)}$ | $0.735_{(0.699, 0.770)}$ | $0.448_{(0.388, 0.515)}$ |
| DAFT (Pölsterl et al., 2021) | | | $0.814_{(0.782, 0.845)}$ | $0.437_{(0.373, 0.522)}$ | $0.730_{(0.694, 0.766)}$ | $0.430_{(0.372, 0.493)}$ |
| Unified (Hayat et al., 2021) | Transformer | ResNet | $0.812_{(0.776, 0.845)}$ | $0.453_{(0.385, 0.533)}$ | $0.719_{(0.681, 0.754)}$ | $0.426_{(0.365, 0.488)}$ |
| MedFuse (Hayat et al., 2022) | | | $0.815_{(0.782, 0.846)}$ | $0.441_{(0.373, 0.520)}$ | $0.728_{(0.692, 0.762)}$ | $0.442_{(0.381, 0.505)}$ |
| DrFuse (Yao et al., 2024) | | | $0.818_{(0.784, 0.850)}$ | $0.460_{(0.391, 0.540)}$ | $0.726_{(0.689, 0.760)}$ | $0.430_{(0.370, 0.495)}$ |
| LSMT (Khader et al., 2023) | | | $0.817_{(0.785, 0.848)}$ | $0.452_{(0.386, 0.535)}$ | $0.722_{(0.688, 0.758)}$ | $0.431_{(0.376, 0.494)}$ |
| Interleaved (Zhang et al., 2023) | | | $0.821_{(0.791, 0.851)}$ | $0.459_{(0.389, 0.539)}$ | $0.721_{(0.683, 0.757)}$ | $0.429_{(0.367, 0.497)}$ |
| CM$^2$ | | | $\mathbf{0.823}_{(0.788, 0.855)}$ | $\mathbf{0.488}_{(0.421, 0.560)}$ | $\mathbf{0.740}_{(0.699, 0.771)}$ | $\mathbf{0.470}_{(0.382, 0.510)}$ |
| MMTM (Joze et al., 2020) | | | $0.813_{(0.778, 0.846)}$ | $0.462_{(0.396, 0.545)}$ | $0.723_{(0.686, 0.761)}$ | $0.435_{(0.380, 0.505)}$ |
| DAFT (Pölsterl et al., 2021) | | | $0.803_{(0.768, 0.836)}$ | $0.432_{(0.363, 0.510)}$ | $0.719_{(0.682, 0.758)}$ | $0.421_{(0.367, 0.486)}$ |
| Unified (Hayat et al., 2021) | Transformer | ViT | $0.812_{(0.778, 0.845)}$ | $0.463_{(0.396, 0.546)}$ | $0.719_{(0.680, 0.753)}$ | $0.412_{(0.353, 0.474)}$ |
| MedFuse (Hayat et al., 2022) | | | $0.818_{(0.786, 0.849)}$ | $0.461_{(0.393, 0.542)}$ | $0.721_{(0.684, 0.759)}$ | $0.431_{(0.371, 0.493)}$ |
| DrFuse (Yao et al., 2024) | | | $0.814_{(0.780, 0.845)}$ | $0.436_{(0.369, 0.516)}$ | $0.717_{(0.680, 0.755)}$ | $0.416_{(0.359, 0.480)}$ |
| LSMT (Khader et al., 2023) | | | $0.815_{(0.784, 0.847)}$ | $0.453_{(0.389, 0.535)}$ | $0.714_{(0.675, 0.751)}$ | $0.424_{(0.365, 0.492)}$ |
| Interleaved (Zhang et al., 2023) | | | $0.818_{(0.786, 0.849)}$ | $0.453_{(0.380, 0.531)}$ | $0.717_{(0.679, 0.753)}$ | $0.433_{(0.371, 0.498)}$ |
| CM$^2$ | | | $\mathbf{0.826}_{(0.790, 0.855)}$ | $\mathbf{0.489}_{(0.422, 0.560)}$ | $\mathbf{0.737}_{(0.700, 0.772)}$ | $\mathbf{0.465}_{(0.394, 0.517)}$ |

*Table 11.* P-values of two-sample bootstrapped $t$-tests of the AUROC and AUPR of CM$^2$ compared to baseline methods. Most of the tests are significant under the 5% significance level.

| Models | IHM | | READM | |
|---|---|---|---|---|
| | AUROC (↑) | AUPR (↑) | AUROC (↑) | AUPR (↑) |
| MMTM (Joze et al., 2020) | 2.02e-06 | 3.55e-180 | 4.40e-100 | 5.36e-291 |
| DAFT (Pölsterl et al., 2021) | 0.1122 | 1.53e-132 | 9.37e-78 | 2.95e-240 |
| Unified (Hayat et al., 2021) | 4.55e-08 | 5.71e-240 | 4.80e-73 | 2.81e-139 |
| MedFuse (Hayat et al., 2022) | 7.73e-07 | 5.66e-129 | 1.11e-92 | 3.69e-173 |
| DrFuse (Yao et al., 2024) | 0.1447 | 4.28e-99 | 6.05e-67 | 6.25e-250 |

