# OpenReview forum: "Cross-Modal Alignment via Variational Copula Modelling"
_ICML.cc/2025/Conference — ICML 2025 poster_

### Official Review · Reviewer_WuxM · 2025-02-26

**Overall Recommendation:** 1

**Summary:**

This paper discusses a multi-modal learning algorithm utilizing copula to "couple" the marginal distributions in each modality. It employs an standard encoder for learning the latent representation for each modality, and model each latent representation as Gaussian mixtures. It then use a copula (selected from a parametric family) to model the joint distribution the modalities, and use mean-field variational inference to optimise the likelihood of the data. Experiments are conducted on single-modal datasets using several single MIMIC modality, and multi-modal datasets using different modalities from MIMIC.

**Claims And Evidence:**

The main claims in this paper is that using copula for the joint distribution of modalities improves performance. This has been supported by experiments on healthcare dataset, namely MIMIC. However, the title of this paper "Cross-Modal Alignment via Variational Copula Modelling" gives the impression that the method proposed is designed for generic tasks. For this title, experiments should be conducted on more broader areas of multi-model learning.

**Essential References Not Discussed:**

In the field of copula-based machine learning methods, there are many recent work which use neural network to represent and learn the copulas. This eliminates the step of manually choose a family of copula and offers better data likelihood. These referenes are entirely missing from the discussion. For example:
1. Ling et al., Deep Archimedean Copulas. NeurIPS 2020.
2. Zhang et al., Deep Copula-Based Survival Analysis for Dependent Censoring with Identifiability Guarantees. AAAI 2024

**Experimental Designs Or Analyses:**

The experimental design is reasonable.

**Methods And Evaluation Criteria:**

The evaluation criteria is area undre the ROC curve and area under precision-recall curve, which are approriate for the problem at hand.

**Other Comments Or Suggestions:**

The paper should be more carefully proofread to avoid minor mistakes. For example:
1. On page 7, line 343, the reference number is not correctly formatted.
2. The $L_{obj}$ in page 4, line 183 should be explicitly provided, at least for the tasks considered in the experiments.

**Other Strengths And Weaknesses:**

Strength: The main strength of this paper is that there are some improvements over the compared methods. However, I am not familiar with the latest literature in multi-modal classification and thus cannot judge if these are significant enough for ICML.

Weakness: The contribution in the aspect of copula learning is rather limited, if any. The theoretical claim in this paper is a reiteration of a well-known (or rather, the most well-known) result in copula theory. The connection of this theorem to the paper is limited.

The qualitative discussion section mainly discusses the characteristics of different families of copulas, e.g., which copula family captures the tail distribution. These are of little relevance to the proposed algorithm's contribution.

**Questions For Authors:**

1. How to ensure that the marginal distributions of the modalities are uniform on the unit interval?
2. For the results reported in the tables, how much different would there be if a different copula family is used?

**Relation To Broader Scientific Literature:**

This paper may contribute to the broader scientific literature with its improved performance of multi-modal learning algorithms.

**Theoretical Claims:**

There is a theoretical claim regarding the "uniqueness of the joint distribution". Unfortunately, the paper simply states the well-known (at least in the field of copula) Sklar's theorem. Citing this theorem here offers almost no insight into the actual multi-modal learning algorithm proposed in the paper. More specifically, the theorem says there exists a unique copula corresponding to the joint distribution; this has no implication on whether the copula learned in this paper is the true underlying copula. In fact, as the copula learning procedure only learns the parameter of a manually selected parametric copula family, it would be unlikely that this learned copula corresponds to the true one.

It would be better if this theoretical claim is removed from the paper.

---

> ### Author Rebuttal · Authors · 2025-04-01
>
> We sincerely thank for your valuable feedback and constructive comments. We take great care in responding to several intriguing discussions raised by you as follows:
> > 1.Applied to generic tasks
>
> Thank you for the suggestion. To support the generality implied by our title, we added results on CMU-MOSI and POM (**see Response 1 to Reviewer NgEi**). CM² consistently outperforms baselines, confirming its effectiveness beyond healthcare.
> > 2.The use of Sklar’s theorem
>
> Thank you for the insightful comment. We use Sklar’s theorem to motivate the decomposition of multimodal joint distributions into marginals and a dependency structure, not as a guarantee of recovering the true copula. While the learned copula is parametric, it enables modeling diverse interaction types (e.g., tail dependencies). To address identifiability, we introduce priors and gradient-preserving sampling, leading to a unique (up to permutation) MAP solution. This theoretical foundation supports robust estimation in complex multimodal settings and connects classical copula theory with scalable deep learning. We will revise the claim to clarify its scope and avoid overstatement.
> > 3.Discuss with copula-based machine learning method
>
> Thank you for the valuable pointers. Our approach leverages parametric families (e.g., Gumbel) to capture domain-relevant properties such as tail dependence. While neural copulas offer flexibility, they introduce additional complexity and may obscure interpretability—key considerations in healthcare. We will include these works in the related discussion.
> > 4.Significance of the result
>
> Thank you for the comment. We performed two-sample bootstrapped t-tests comparing CM² with baselines; most results show statistically significant gains. Please see **Response 2 to Reviewer NgEi** for p-values supporting the improvements.
> > 5.The connection of this theorem to the pape
>
> Thank you for the comment. While Sklar’s theorem is classical, our contribution lies in operationalizing it within a scalable deep multimodal framework. We leverage its decomposition to decouple marginals (modeled via GMMs) from dependencies (via parametric copulas), enabling feature-level alignment under missing modalities. Our framework also addresses challenges like marginal non-identifiability and tail dependence through variational inference and gradient-preserving sampling. This structured integration of copula theory into end-to-end multimodal learning constitutes a novel and practical contribution beyond simply restating the theorem.
> > 6.Tail distribution of diffirent copula family
>
> We appreciate the reviewer’s point. The qualitative discussion of copula families is relevant because it highlights how different dependency structures—especially tail behaviors—affect the joint modeling process. In domains like healthcare, modeling extreme events is critical for risk-sensitive tasks. This analysis also guides practitioners in selecting appropriate copula families based on data characteristics, enhancing the interpretability and adaptability of our method.
> > 7.How to ensure that the marginal distributions of the modalities are uniform on the unit interval
>
> We ensure uniform marginals via the probability integral transform. In practice, for each modality we first model the latent feature distribution (typically using a flexible Gaussian mixture model) and then compute its cumulative distribution function (CDF). By transforming the latent variables through their respective CDFs, we obtain variables that are uniformly distributed over [0, 1], which is guaranteed by the properties of the CDF. This transformation is a core component of our framework, aligning the modality-specific representations to a common uniform scale and facilitating the subsequent copula-based joint modeling.
> > 8.How much different would there be if a different copula family is used
>
> Due to the limited number of observations, the performance across copula families tends to be more variable in the matched subset . On the other hand, the observations are more sufficient in the partially matched datasets, leading to relatively stable performance across families. This demonstrates the importance of choosing a correct copula family since the tail risks is more evident as the number of observations decreases.
> > 9.Explicit form of $ L_{obj}$
>
> The overall objective loss in our framework is defined as $L_{obj} = L_{task} + \lambda_{cop} \cdot L_{\text{copula}}$, where $L_{task}$ is the task-specific loss (such as cross-entropy for classification tasks) and $L_{\text{copula}}$ is the negative log-likelihood of the joint copula model, with $\lambda_{cop}$ balancing the two terms. We would proofread to correct the notational errors in future versions of the manuscript.
> >10.Typos
>
> Thanks for pointing out the rendering error. We will fix this in the final version.

---

### Official Review · Reviewer_g4J9 · 2025-03-05

**Overall Recommendation:** 3

**Summary:**

The paper presents a multimodal learning framework based on Copula theory. The modalities are modeled using a Gaussian mixture distribution, and a joint copula model is applied to the joint distribution. The proposed method is validated on a healthcare dataset, considering both cases where modalities are missing and where all modalities are present.

**Claims And Evidence:**

The claims are supported by evidence however the paper could benifit from additional discussions (see questions).

**Essential References Not Discussed:**

Not that I'm aware of.

**Experimental Designs Or Analyses:**

The experimental design would benefit from validation on datasets from other domains beyond healthcare.

**Methods And Evaluation Criteria:**

The experiments are well-structured and make sense.

**Other Comments Or Suggestions:**

Broken reference in line 344.

**Other Strengths And Weaknesses:**

Strengths:

- The paper is well-written and easy to read.
- The method introduces copula theory to multimodal learning, which is an interesting approach.
- The experimental validation demonstrates the method's performance compared to state-of-the-art techniques.

Weaknesses:

- The theoretical guarantees are not clearly stated.
- The paper would benefit from validation on datasets beyond healthcare.

**Questions For Authors:**

1-  Can you discuss how initial marginal modeling (e.g., via GMM) might introduce biases or errors that impact the joint copula estimation.

2- Can the method be generalized beyond healthcare and performs well in other domains.

3 - Can you discuss the scalability to higher numbers of modalities (beyond bimodal/trimodal).

4 - How does this method compare to unsupervised approaches like multimodal VAEs (see some references below) when labels are not available? The advantage of these methods is that they do not require labeled datasets to learn meaningful representations.

Sutter, T. M., Daunhawer, I., & Vogt, J. E. (2021). Generalized multimodal ELBO. arXiv preprint arXiv:2105.02470.
Hwang, H., Kim, G. H., Hong, S., & Kim, K. E. (2021). Multi-view representation learning via total correlation objective. Advances in Neural Information Processing Systems, 34, 12194-12207.


**Post-rebuttal comment**

The rebuttal has addressed my questions, and I have decided to maintain my initial positive score.

**Relation To Broader Scientific Literature:**

The paper offers a new perspective on multimodal learning using copulas, which is interesting, especially given that multimodal learning is a broad and versatile domain.

**Theoretical Claims:**

The author uses Sklar’s theorem to demonstrate the uniqueness of the copula joint distribution. While this is valid in this context, it should be stated more clearly how the initial modeling of the marginals via GMM could impact this claim.

---

> ### Author Rebuttal · Authors · 2025-04-01
>
> We sincerely thank for your valuable feedback and constructive comments. We take great care in responding to several intriguing discussions raised by you as follows:
> > 1.Impact of initial modeling of the marginals via GMM
>
> Thank you for the insightful comment. Our initial marginal modeling, which assumes a feature-level representation, is inherently task-agnostic and thus universally applicable across different downstream tasks. By modeling each modality with a Gaussian mixture model—the most flexible and expressive assumption available in latent space - the framework can robustly capture complex, multimodal distributions. Although any marginal estimation could introduce potential biases, our joint optimization via the ELBO ensures that even minor discrepancies are corrected during copula estimation, allowing the dependency structure between modalities to be accurately aligned.
> > 2.Validation on datasets from other domains
>
> To evaluate generalizability beyond healthcare, we conducted additional experiments on CMU-MOSI and POM. CM² achieves the best performance across all metrics (see table below) compared to other methods, demonstrating its broad applicability. We will include these results in the final version.
> |Model|**CMU-MOSI**|**CMU-MOSI**|**CMU-MOSI**|**POM**|**POM**|**POM**|
> |-|-|-|-|-|-|-|
> ||MAE|Accuracy|F1|MAE|Corr|Accuracy|
> |Unified|1.21|0.656|0.657|0.862|0.213|0.353|
> |MedFuse|1.11|0.700|0.696|0.861|0.262|0.334|
> |DrFuse|1.12|0.700|0.700|0.869|0.243|0.338|
> |LMF|1.13|0.697|0.698|0.856|0.266|0.343|
> |TFN|1.18|0.682|0.682|0.858|0.263|0.358|
> |**CM²**|**1.08**|**0.710**|**0.708**|**0.840**|**0.281**|**0.365**|
> > 3.Theoretical guarantees
>
> We acknowledge that our current theoretical guarantee, primarily rooted in classical results such as Sklar’s theorem, may appear limited in scope. However, our primary focus in this work is methodological and applied, targeting the complex challenges of healthcare multimodal data. Our main contribution lies in demonstrating the practical efficacy of integrating copula-based alignment with flexible Gaussian mixture modeling in real-world settings, which is supported by strong empirical results. We view our current theoretical framework as a solid foundation and plan to actively explore deeper theoretical insights—such as more rigorous guarantees on the learned dependency structure—in future work.
> > 4.The scalability to higher numbers of modalities
>
> We assume a fully connected density model where the multivariate Gumbel copula can be obtained by the Archimedean copula
> $$c(\mathbf{u}) = \psi^{(d)}(t(\mathbf{u})) \prod_{j=1}^d (\psi^{-1})'(u_j)$$
> where $\varphi(t;\alpha) = (\log t)^\alpha$ for the Gumbel copula Hence the higher number of modalities (i.e., when $M > 3$) can be handled in this case
> > 5.Compare to unsupervised approaches
>
> We compared CM² against unsupervised multimodal VAEs including MoPoE-VAE and MVTCAE on MIMIC4. While these methods excel in synthetic and vision datasets, they are not tailored for downstream predictive tasks. In contrast, CM² integrates copula-based alignment with task supervision, capturing fine-grained dependencies (e.g., tail risks) that are critical in healthcare. As shown in tables below, CM² consistently outperforms these methods on both fully and partially matched IHM/READM settings. We attribute this to its stronger alignment of modality-specific representations under distributional assumptions that go beyond standard VAE objectives.
>
> **Totally Matched**
> |Model|**IHM AUROC**|**IHM AUPR**|**READM AUROC**|**READM AUPR**|
> |-|-|-|-|-|
> |MVTCAE|0.736|0.341|0.678|0.362|
> |MoPoE-VAE|0.730|0.338|0.671|0.357|
> |**CM²**|**0.827**|**0.492**|**0.737**|**0.466**|
>
> **Partially Matched**
> |Model|**IHM AUROC (↑)**|**IHM AUPR (↑)**|**READM AUROC (↑)**|**READM AUPR (↑)**|
> |-|-|-|-|-|
> |MVTCAE|0.767|0.347|0.701|0.366|
> |MoPoE-VAE|0.778|0.368|0.709|0.379|
> |**CM²**|**0.858**|**0.527**|**0.771**|**0.486**|
>
> > 6.Typos
>
> Thanks for pointing out the rendering error. We will fix it in the final version.

---

> > ### Comment · Reviewer_g4J9 · 2025-04-02
> >
> > I thank the authors for their response. The additional answers addressed my concerns. Therefore, I have decided to maintain my initial positive score.

---

### Official Review · Reviewer_kwzw · 2025-03-12

**Overall Recommendation:** 3

**Summary:**

This work primarily focuses on the problem of multimodal supervised learning, where some modalities may be missing. The authors model the joint latent distribution of all modalities using a copula model with finite Gaussian mixture marginals. In the presence of missing modalities, they impute the missing latents by generating samples from the copula model conditioned on the available modalities.

**Claims And Evidence:**

I'm not aware of unsupported claims, though I do have questions regarding the expt setting and methodologies.

**Essential References Not Discussed:**

Several recent works on multimodal matching and alignment are not discussed, such as *Propensity Score Alignment of Unpaired Multimodal Data* and *Unpaired Multi-Domain Causal Representation Learning*. I would appreciate a discussion on whether these methods could be applied to the modality alignment tasks presented in this work, as well as additional experiments benchmarking against these approaches.

Additionally, it is worth noting that *Propensity Score Alignment of Unpaired Multimodal Data* also addresses the task of missing modality imputation, which should be mentioned and compared where relevant.

**Experimental Designs Or Analyses:**

see other comments.

**Methods And Evaluation Criteria:**

- It might be due to my lack of knowledge, but it is unclear which specific modality alignment tasks are reported. Additionally, how are these alignment losses computed? Perhaps I missed it, but I could not find a precise description in the main text.

- Most experiments in this work focus on performing supervised inference with missing modalities rather than analyzing the dependence structure across different modalities. Given this, I feel the title of the work could potentially be misleading.

- Selecting an appropriate copula family is crucial, as different families exhibit distinct dependence properties. For example, the Gaussian copula lacks extreme tail dependence, whereas the Gumbel copula does. Could you elaborate on why tail dependence is relevant to your application and why you specifically chose the Gumbel copula? Additionally, how does tail dependence correspond to "the strongest signals" in each modality? Could you clarify this statement with precise reasoning?

- Could you describe in detail how imputation for missing latents is performed? I assume the process involves first generating a uniform random variable from the copula function conditioned on the available modalities, followed by applying the inverse CDF of the Gaussian mixture to recover  z. Is this correct?

- How do you specify the dependence structure in your multivariate copula function when dealing with more than two modalities? Do you adopt a vine copula structure? If so, how do you determine the vine structure for each problem? Alternatively, if you assume a fully connected density model between all modalities, this would introduce quadratic complexity in imputing missing modalities, which may become impractical with a large number of modalities. Could you clarify your approach?

**Other Comments Or Suggestions:**

A few minor points:

-  I find calling the objective an ELBO somewhat misleading, as it is unclear how it serves as a lower bound for the evidence $\log p(x_1, .., x_m)$. This is not a VAE objective (which is typically unsupervised); rather, it appears to be a linear combination of the log-likelihood of the latent distribution and a supervised machine learning objective. Alternatively, if you are referring to the evidence $ \log p(y)$, for mathematical rigor, it would be beneficial to explicitly write out the probabilistic model for $y|x_1, ..., x_M$ for clarification.


- Latex rendering error in line 344.

**Other Strengths And Weaknesses:**

see other comments.

**Questions For Authors:**

see other comments.

**Relation To Broader Scientific Literature:**

n/a

**Theoretical Claims:**

n/a

---

> ### Author Rebuttal · Authors · 2025-04-01
>
> We sincerely thank for your valuable feedback and constructive comments. We take great care in responding to several intriguing discussions raised by you as follows:
> > 1.Modality Alignment Tasks & Alignment Losses
>
> In our framework, modality alignment is achieved through the copula loss, which explicitly models and optimizes the dependency structure among modalities in a shared latent space. While we do not define separate alignment tasks per se, we evaluate the effectiveness of alignment by measuring cross-modal prediction performance—i.e., how well one modality can predict or reconstruct another. This serves as an implicit test of alignment quality. The copula loss, computed as the negative log-likelihood of the joint copula distribution, encourages aligned representations by capturing inter-modal dependencies beyond marginal distributions. We will clarify this connection more explicitly in the main text.
>
> > 2.Title of the Work
>
> Thank you for your suggestions. We would revised the title in future versions of the manuscript to better reflect the contents of our work.
>
> > 3.Tail dependence and Copola Family Analysis
>
> Tail dependence is critical in our application because it captures the co-occurrence of extreme events across modalities—events that are often indicative of critical outcomes in healthcare, such as severe physiological deterioration or acute anomalies in imaging. We specifically chose the Gumbel copula because its ability to model strong upper tail dependence aligns with our goal of highlighting these "strongest signals" present in the extreme ends of each modality's distribution; these signals, which correspond to rare yet significant observations, are essential for accurate risk prediction and decision-making. In contrast, a Gaussian copula would underrepresent these tail dependencies, potentially diluting the impact of extreme but clinically meaningful observations.
>
> > 4.How imputation for missing latents is performed
>
> Yes, your assumption is correct—thank you for the accurate summary.
> > 5.Specify the dependence structure dealing with more than two modalities
>
> In our current framework, we specify the dependence structure using symmetric copulas rather than adopting a vine copula structure. Specifically, we assume a fully connected density model where the multivariate Gumbel copula is given by
> $$C(u_1, \dots, u_M) = \exp\{-[(-\log u_1)^\theta + \cdots + (-\log u_M)^\theta ]^{1/\theta}\}$$
> and a general Archimedean copula is defined as
> $$C(u_1, \dots, u_M) = \phi^{-1}\left( \phi(u_1) + \cdots + \phi(u_M) \right),$$
> where $\phi$ is the generator function and $\theta$ controls the dependence strength. This symmetric, feature-level assumption makes our approach task-agnostic and computationally tractable, avoiding the quadratic complexity that a vine copula structure would introduce when imputing missing modalities. We plan to investigate more complex dependence models, such as vine copulas with tailored structure selection, in future work.
>
> > 6.Discuss on Recent Multimodal Matching Work
>
> We both adopt probabilistic assumptions on latent features—while Propensity Score Alignment (PSA) models intra-modal distributions from a causal perspective, our method leverages copula theory to model inter-modal dependencies, particularly capturing complex interactions such as tail dependence. We compare our method with PSA on MIMIC4 in the table below. Our method outperforms PSA in both matched and partially matched settings.
>
> **Totally Matched**
> |Model|**IHM AUROC**|**IHM AUPR**|**READM AUROC**|**READM AUPR**|
> |-|-|-|-|-|
> |PSA|0.744|0.346|0.692|0.370|
> |**CM²**|**0.827**|**0.492**|**0.737**|**0.466**|
>
> **Partially Matched**
> |Model|**IHM AUROC**|**IHM AUPR**|**READM AUROC**|**READM AUPR**|
> |-|-|-|-|-|
> |PSA|0.792|0.386|0.720|0.391|
> |**CM²**|**0.858**|**0.527**|**0.771**|**0.486**|
>
> > 7.Calling of ELBO
>
> We use the term "ELBO" in a generalized sense to reflect a variational inference framework that combines both supervised and unsupervised objectives. Specifically, our method models the joint latent distribution via a copula and optimizes a variational lower bound that includes the copula log-likelihood (unsupervised) and a task-specific loss. For classification tasks, this supervised term corresponds to the KL divergence between the predictive label distribution and the true label (i.e., a variational form of cross-entropy), which aligns with the evidence lower bound on $\log p(y)$ under a probabilistic decoder. For unsupervised tasks like clustering, this component can instead represent a metric-based loss (e.g., intra-cluster distances). Thus, our objective remains variational in nature, enabling principled integration of both labeled and unlabeled information, and we will clarify this probabilistic interpretation of $p(y|x_1, \dots, x_M)$ more explicitly for mathematical rigor.
>
> > 8.Typos
>
> Thanks for pointing out the rendering error. We will fix it in the final version.

---

> > ### Comment · Reviewer_kwzw · 2025-04-06
> >
> > Thank you for your clarification. Given the additional experiements, I'm willing to raise the score by 1.

---

### Official Review · Reviewer_NgEi · 2025-03-14

**Overall Recommendation:** 3

**Summary:**

The paper proposed a copula modeling method for multi-modal representation learning, which could model the interactions between modalities and impute the missing modalities through sampling from learned marginals. The method was empirically evaluated on healthcare benchmarks MIMIC-III and MIMIC-IV datasets for two classification tasks. Code is provided in anonymous github.

**Claims And Evidence:**

Yes. Most claims are well supported by literature and/or experiments results, though some evidence (e.g. significance test) could only be found in the supplementary.

**Essential References Not Discussed:**

No.

**Experimental Designs Or Analyses:**

Yes.
- Minor issues on experimental designs, especially on the choice of encoders for baseline methods. It seems some baseline methods and/or tasks are sensitive to the choice of encoders. Should the experiment design allow optimizing choice of encoders given a baseline method? Instead of using the same encoders for all baselines.

**Methods And Evaluation Criteria:**

Yes. The proposed methods are evaluated on two classification tasks from MIMIC datasets.
- From the results table (e.g. Table 2 and Table 3), it seems the AUROC performances are close to each other (e.g. $CM^2$ and DrFuse). It might be intuitive to compare the ROC curves for better interpretation.
- As mentioned in the limitations and future works, other types of multi-modal datasets are needed to prove the utility of the proposed method beyond healthcare datasets.

**Other Comments Or Suggestions:**

- wrong reference of tables in "ablation on different families of Copula" in section 4.4 on page 7. "Table 11" might be Table 6?
- Are those measures in Table 5 and 6 statistically significant?

**Other Strengths And Weaknesses:**

Strength:
- The paper shows decent technical soundness (if considering some experiment results in the supplementary materials). Results, though not cross-validated, are tested with bootstrapping experiments. Confidence intervals are reported for most performance metrics, except table 5 ablation study and table 6 different copula families.

Weakness:
- Some of the important experiment design/results should be included in the main text, e.g. significance tests in table 11.  Performances with significant p-values could be marked with underline, or star, etc, in table 2 and 3. Otherwise the CIs seem to be non-significant at the first glance.
- The discussion of experiment results lack the in-depth analysis with regard to the healthcare context. Though the overall AUROC of the proposed method outperforms baselines, the predictions of individual cases might vary across baselines. It'll be better to analyze how much agreement/disagreement of individual cases predictions from different baselines, and what are the potential interpretation and pros/cons of the proposed method.

**Questions For Authors:**

- How will the proposed method deal with categorical features, where Gaussian distribution cannot be assumed?
- How will the proposed method be impacted by potential conflicts among different modalities, either due to data errors or by nature?

**Relation To Broader Scientific Literature:**

The paper might provider an inspiring idea on how to align multi-modal distributions through copula modeling.

**Theoretical Claims:**

No. The Sklar's Theorem in part 3.5 is well established.

---

> ### Author Rebuttal · Authors · 2025-04-01
>
> We sincerely thank for your valuable feedback and constructive comments. We take great care in responding to several intriguing discussions raised by you as follows:
> > 1.Results on other types of multi-modal datasets
>
> Thank you for the suggestion. To evaluate generalizability beyond healthcare, we conducted additional experiments on CMU-MOSI and POM. CM² achieves the best performance across all metrics (see table below) compared to other methods, demonstrating its broad applicability. We will include these results in the final version.
> |Model|**CMU-MOSI**|**CMU-MOSI**|**CMU-MOSI**|**POM**|**POM**|**POM**|
> |-|-|-|-|-|-|-|
> ||MAE|Accuracy|F1|MAE|Corr|Accuracy|
> |Unified|1.21|0.656|0.657|0.862|0.213|0.353|
> |MedFuse|1.11|0.700|0.696|0.861|0.262|0.334|
> |DrFuse|1.12|0.700|0.700|0.869|0.243|0.338|
> |LMF|1.13|0.697|0.698|0.856|0.266|0.343|
> |TFN|1.18|0.682|0.682|0.858|0.263|0.358|
> |**CM²**|**1.08**|**0.710**|**0.708**|**0.840**|**0.281**|**0.365**|
> > 2.Significance of the result
>
> We thank the reviewer for the thoughtful suggestions. We performed two-sample bootstrapped t-tests to compare CM² against **SOTA baselines** and **ablations**. Most comparisons yielded significant p-values, as shown in table below. We will revise Tables 2 and 3 by marking results with significant differences (e.g., using *) for better clarity. Due to space limits, we were unable to include the ROC curves now but will include them in the final version for improved visual interpretation.
> |Model|**IHM AUROC**|**IHM AUPR**|**READM AUROC**|**READM AUPR**|
> |-|-|-|-|-|
> |MIMIC-III Paired|3.95e-46|0.321|0.002|0.166|
> |MIMIC-IV Paired|1.47e-09|4.17e-19|1.34e-11|4.73e-33|
> |MIMIC-III Partial|4.02e-11|8.93e-32|0.290|0.003|
> |MIMIC-IV Partial|0.1447|4.28e-99|6.05e-67|6.25e-250|
>
> |Model|Matched|**IHM AUROC**|**IHM AUPR**|**READM AUROC**|**READM AUPR**|
> |-|-|-|-|-|-|
> |w/o Copula alignment|✗|0.001|1.73e-30|2.00e-93|3.47e-62|
> |w/o GPS|✗|0.398|4.83e-4|1.02e-22|7.07e-17|
> |w/o fusion module|✗|0.003|0.029|4.15e-28|7.09e-11|
> |w/o Copula alignment|✓|5.33e-34|9.48e-68|6.86e-35|4.79e-48|
> |w/o fusion module|✓|5.46e-26|1.64e-46|4.95e-26|4.02e-50|
> > 3.Choice of encoders
>
> We appreciate the reviewer’s concern. While we followed encoder settings from original papers (e.g., MedFuse, DrFuse), we additionally tested each method with different encoders in **Table 10 in Appendix**. CM² consistently outperforms all baselines across these settings, confirming its robustness to encoder choices. We will clarify this experimental detail in the final version of the paper.
> > 4.In-depth analysis with regard to the healthcare context
>
> We appreciate this insightful suggestion. Our method is designed to optimize global performance metrics through robust copula-based multimodal alignment, and it inherently relies on probabilistic inference and sampling, which introduce variability at the individual case level. In our framework, each prediction is generated as an expectation over a learned latent distribution that is optimized to maximize overall AUROC and AUPR, rather than to provide deterministic case-specific outputs. This stochastic nature—especially under missing modality scenarios—limits the reliability of direct one-to-one comparisons of individual case predictions across baselines.
>
> > 5.Deal with categorical features
>
> Thank you for the question. Our GMM assumption applies to latent representations, not raw inputs. Categorical features are embedded and transformed before modelling; thus, the Gaussian assumption holds in the latent space.
>
> > 6.Impact by potential conflicts among different modalities
>
> We appreciate the reviewer’s point. Our probabilistic framework regularizes modality interactions via distributional assumptions. This helps mitigate the impact of noisy or conflicting modalities by reducing sensitivity to outliers in the joint space.
>
> > 7.Typos
>
> Thank you for catching this. The reference to Table 11 in Section 4.4 should be corrected to Table 6. We will fix this and another rendering error in the final version.

---

> > ### Comment · Reviewer_NgEi · 2025-04-04
> >
> > I appreciate the authors' detailed response with additional experiment results to address my concerns.

---

### Decision · Program_Chairs · 2025-05-01

**Decision:**

Accept (poster)

**Comment:**

This manuscript presents CM², a copula-based framework for multimodal representation learning that models interactions between modalities and handles missing modality imputation through learned marginal sampling. The work demonstrates strong empirical performance on healthcare benchmarks MIMIC-III/IV and additional validation on CMU-MOSI and POM datasets. The reviewers highlighted several strengths, including the paper's technical soundness supported by comprehensive statistical testing, the broad applicability of the copula-based approach beyond healthcare domains, and the practical utility of capturing tail dependencies for risk-sensitive tasks. The rebuttal successfully addressed initial concerns about generalizability by providing additional experimental results on non-healthcare datasets. Some limitations noted by reviewers included the need for more theoretical guarantees beyond Sklar's theorem and deeper analysis of individual case predictions in healthcare contexts. The authors' thorough response clarified the methodological choices, particularly regarding copula family selection and marginal distribution modeling. After the rebuttal, reviewers maintained or improved their initial positive assessments, with scores ranging from weak accept to accept, appreciating the additional experiments and clarifications provided. The consensus highlighted the work's methodological contribution in integrating copula theory with deep multimodal learning, while acknowledging areas for future theoretical development.